# A neotropical perspective on the uniqueness of the Holocene among interglacials

J. Schiferl[1], M. Kingston[1], C. M. Åkesson[1], B. G. Valencia [2], A. Rozas-Davila[1], D. McGee [3], A. Woods[4], C. Y. Chen[5], R. G. Hatfield [6], D. T. Rodbell [7], M. B. Abbott [4] & M. B. Bush [1] ✉

Understanding how tropical systems have responded to large-scale climate change, such as glacial-interglacial oscillations, and how human impacts have altered those responses is key to current and future ecology. A sedimentary record recovered from Lake Junín, in the Peruvian Andes (4085 m elevation) spans the last 670,000 years and represents the longest continuous and empirically-dated record of tropical vegetation change to date. Spanning seven glacial-interglacial oscillations, fossil pollen and charcoal recovered from the core showed the general dominance of grasslands, although during the warmest times some Andean forest trees grew above their modern limits near the lake. Fire was very rare until the last 12,000 years, when humans were in the landscape. Here we show that, due to human activity, our present interglacial, the Holocene, has a distinctive vegetation composition and eco-logical trajectory compared with six previous interglacials. Our data reinforce the view that modern vegetation assemblages of high Andean grasslands and the presence of a defined tree line are aspects of a human-modified landscape.

In the tropical Andes, glacial-interglacial cycles induced vertical migrations of plants of c. 1000–1500 m[1,2]. As glacials ended, taxa migrated upslope[3–5], responding to rising temperatures. Two long paleoecological records for the high Andes show contrasting inter-glacial histories. In the Lake Fúquene record from the forested setting of the high plains around Bogotá (5° N, 2540 m elevation), Colombia, cold grasslands replaced forests during glacial periods, but each interglacial was marked by a temperature-driven upslope migration of forest species to produce assemblages like the forests of today[6] (Fig. 1). Conditions during interglacials appear to have been warm and wet. At Lake Titicaca (18°S, 3810 m elevation) in the Bolivian Altiplano, a temperature driven response of an upslope migration of forest was interrupted by aridity during two major interglacials, those of Marine Isotope Stages (MIS) 5e and 9. During both events, Andean forest migration stalled as the system transitioned to a saltmarsh, suggesting

a drought-driven state[7]. Consequently, these two sites show opposing patterns of when droughts peak, with the driest times in Colombia being during glacials versus interglacials at Titicaca.

Bradbury[8] suggested exactly this kind of climatic hinge point in the Andes, with locations north and south of c. 10°S having opposing precipitation responses to glacial-interglacial cycles. At 11°S, lying between Bolivia and Colombia, Lake Junín Peru, allows an investigation of the effects of interglacials of the last 700,000 years on vegetation composition to evaluate if drought or temperature had the strongest effects.

Past interglacials may offer insights as analogs for our warmer-than-modern future[9]. The orbital parameters of interglacials within the last 700,000 years indicate that MIS 11 was most similar to MIS 1 (the Holocene)[10]. As MIS 11 had natural fire regimes, a full complement of megaherbivores, and seasonality similar to that of our present

[1]Institute for Global Ecology, Florida Institute of Technology, Melbourne, FL 32901, USA. [2]Facultad de Ciencias de La Tierra y Agua, Universidad Regional Amazónica Ikiam, Tena, Ecuador. [3]Department of Earth, Atmospheric and Planetary Sciences, Massachusetts Institute of Technology, Cambridge, MA 02139, USA. [4]Department of Geology and Environmental Science, University of Pittsburgh, Pittsburgh, PA, USA. [5]Chemical and Isotopic Signatures Group, Nuclear and Chemical Sciences Division, Lawrence Livermore National Laboratory, Livermore, CA 94550, USA. [6]Department of Geological Sciences, University of Florida, Gainesville, FL 32611, USA. [7]Geoscience Department, Union College, Schenectady, NY 12308, USA. ✉e-mail: mbush@fit.edu

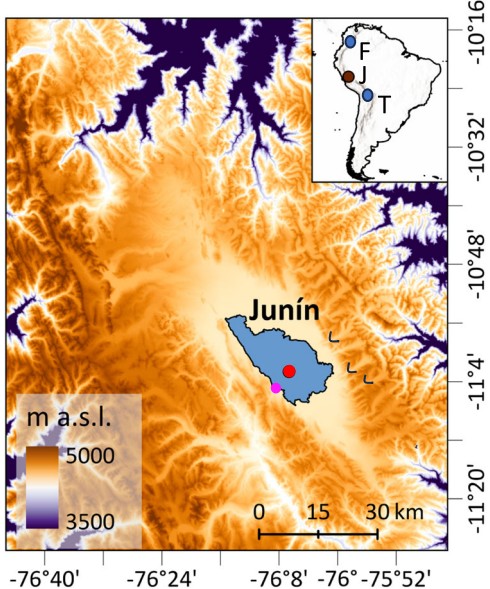

**Fig. 1 | Map showing the location of Lake Junín relative to other sites mentioned in text and the 2015 coring location (red circle).** The pink circle denotes the sediment core raised in 1996[56]. Black arcuate lines are approximate extent of glaciers during Marine Isotope Stages 2 and 3; downvalley topographic ridges are pre-Marine Isotope Stage 3 moraines[78]. Inset map: circles indicate Andean records discussed in the text: J Junín, F Fúquene, High Plain of Bogotá[33], T Lake Titicaca[7]. Maps are derived from NASA Shuttle Radar Topography Mission and mapped in Esri ArcGIS Pro.

interglacial, it may offer insights into when the scale of alteration of habitats in the only interglacial occupied by humans exceeded natural variability in MIS 1.

Human modification of Andean ecosystems began in the terminal Pleistocene as fire frequencies increased markedly and megafauna were functionally eradicated by c. 12.5 ka[11–14]. Subsequent camelid domestication, crop cultivation, and burning have transformed Andean landscapes to such a point that montane grasslands became a manufactured landscape[15]. Pre-human punas may have been richer in woody taxa than those of today[16,17] and had a softer boundary with Andean tree lines (the upper boundary of continuous Andean forest cover)[18]. Here we can investigate these ideas across numerous glacial-interglacial transitions.

The impact of human activity on Andean tree lines has been discussed extensively[19–21]. The expectation is that by burning highland grasslands, fires would have eroded the upper edge of the tree-line causing it to shift downslope[22]. Similarly, livestock grazing and management to maintain dense grass cover, which inhibits tree seedlings from establishing, could all contribute to preventing forests reaching their full potential to colonize grasslands[21]. Consequently, it has been suggested that potential natural tree lines might lie above their modern range of c. 3400–3700 m elevation[23]. Prior paleoecological studies have attempted to track tree lines using fossil pollen, but evidence for tree line migration during the late Holocene found either no change[24] or a c. 200 m downslope displacement[25]. An investigation of forest cover during MIS 11 could provide new insights into this longstanding question.

Although lying at 4085 m elevation above sea-level, Lake Junín was not glaciated during the last seven interglacials[26] (Fig. S1). In 2015, an 88 m-long core that spanned the last 670 ka, was raised from 12 m water depth in Lake Junín (11°01′S;76°07′W, 4085 m asl), Peru (Fig. 1). A robust chronology for the core was developed using [14]C and U-Th dating paired with paleomagnetic data[27–29]. Nine tie-points were used

to make minor adjustments to improve alignment with the EPICA dome C ice core from Antarctica[26] (Fig. S2).

A fossil pollen and charcoal analysis of the Junín core, JUN 15, yielded a paleoecological history in which fire occurrence and vegetation composition varied substantially through time. Through seven glacial-interglacial cycles, the area around the lake oscillated between glacial foreland and grassland, with varying amounts of Andean woodland during the warmest periods. In this study, we show how human activity altered the trajectory of ecosystems, making our modern interglacial ecologically unique.

## Results and discussion

### Assemblage changes between glacials and interglacials

The fossil pollen data from core JUN 15 indicate that the area surrounding Lake Junín was always a grassland, and Poaceae (grasses) remained the most abundant pollen type in almost all glacial and interglacial samples. As is typical of large Andean lakes[20], pollen influx to Lake Junín was low but highly variable (Fig. 2). Peaks of pollen influx occurred during interglacial events. As the environment warmed, the density of plants went from scattered individuals to a dense sward, and the landscape became more productive and plant diversity increased (Fig. S3). Poaceae pollen influx (grains per $cm^2$ per yr) rose by two orders of magnitude as landscape productivity increased during interglacials.

As would be expected from many other pollen records, interglacial warming caused an upslope migration of species that brought montane forests closer to the lake[30]. *Podocarpus, Hedyosmum*, and *Weinmannia*, trees of modern upper Andean forests, occurred at such abundances that they probably grew close to the lake (Fig. S3). Although these taxa are typical of upper Andean forest, many other Andean pollen types that often co-occur with them such as *Bocconia*, Myrtaceae and Rubiaceae, were not documented in these samples. Yet other arboreal taxa, such as *Vallea, Escallonia*, and *Myrsine*, were far less abundant than is usual in upper montane forest samples. Taken together, these unusual abundances suggest that these habitats of peak warmth were without modern analog sensu[31].

During glacials, in addition to Poaceae, members of the Puna shrubland (Table S1), primarily comprised of Asteraceae and *Polylepis* (Fig. 3), were important components of the pollen spectra. Although most abundant during glacials, puna shrubland taxa also occurred during cool or moist periods within interglacials. *Polylepis* is a small tree that can be found up to the modern ice-limit[32]. Its abundance during glacials, at c. 10% of the pollen sum, probably reflected its presence near the lake (Fig. 3). The very low pollen influx during glacials, however, indicated that rather than growing as a woodland, the *Polylepis* probably survived as scattered clumps in favorable microrefugia near the lake.

Another arboreal taxon, *Alnus*, was also most abundant during glacials (Fig. 3). *Alnus*, however, probably did not grow locally. A pioneer tree of disturbed forest edges, *Alnus* pollen is well known to be massively overrepresented in pollen spectra from glacial forelands through being blown upslope[33], reaching as much as 400% of the pollen sum in some montane grasslands[34]. That *Alnus* had very high relative abundances at Junín during glacials (30–45%), but its representation fell to <1% during interglacials is entirely consistent with this pollen coming from trees growing at a lower elevation. Following Hooghiemstra[33] we excluded *Alnus* from our Andean forest subtotals.

Lower montane forest taxa also peaked in abundance during times of low pollen influx (Fig. 2). This group was diverse, but included taxa that today would generally be found below 2800 m elevation, e.g. Moraceae-Urticaceae, *Celtis, Trema*, and *Cecropia*. Continuously updrafted by rising parcels of air, these pollen grains were transported many kilometers and only form a significant percentile component of the pollen flora when local pollen production was low.

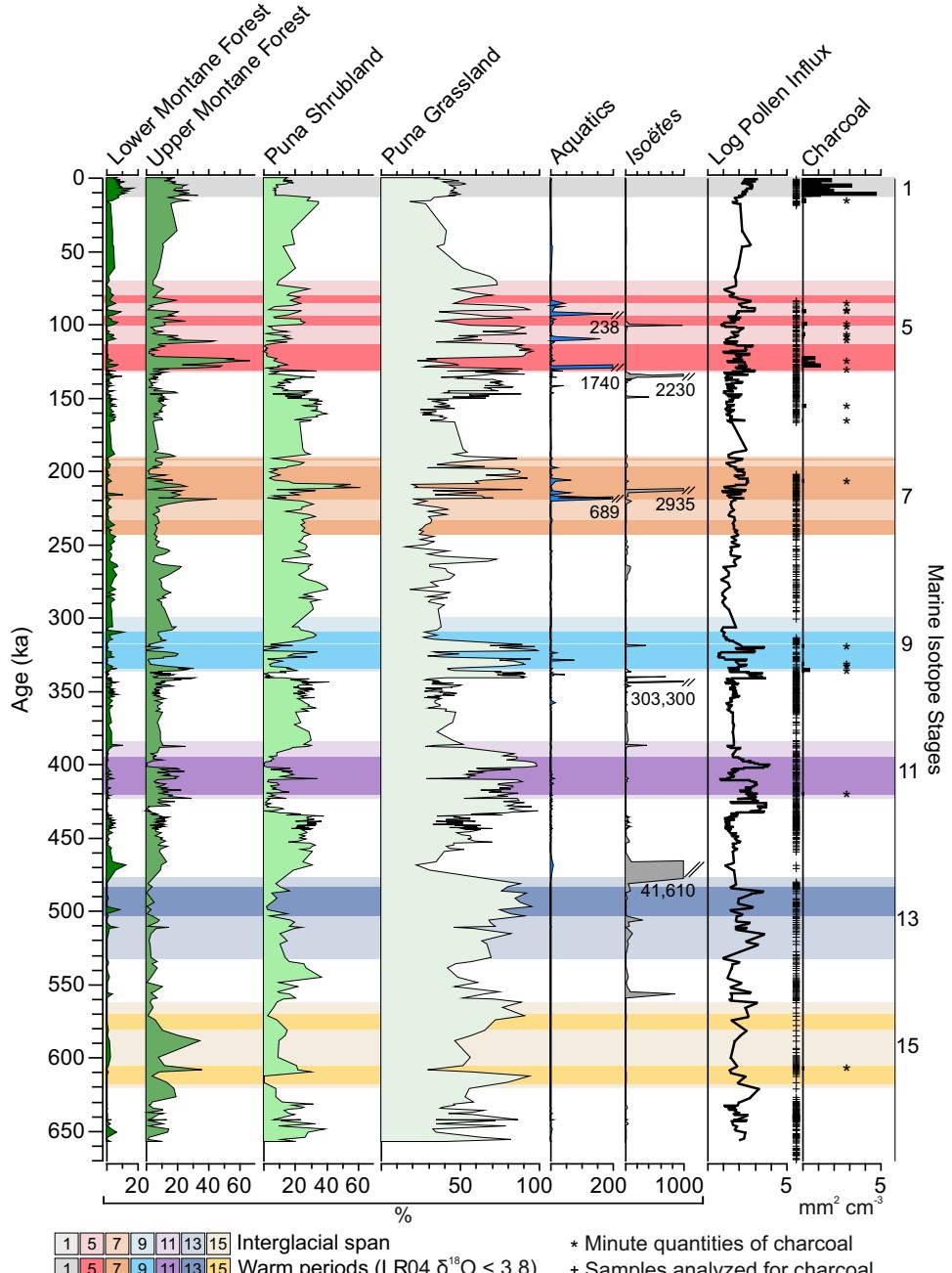

**Fig. 2 | Habitat representation around Lake Junín, Peru, based on fossil pollen recovered from core JUN 15.** For taxa assigned to each habitat see Table S1. Aquatic plants, Isöetes and *Alnus* are excluded from the pollen sum. Source data are provided as a Source Data file.

The spores of *Isoëtes*, an aquatic quillwort, exhibited isolated, massive, peaks of abundance that reached 303,300% of the terrestrial pollen sum at 327 ka and 41,610% at 464 ka (Fig. 2). *Isoëtes* would have grown in the shallows of the lake margin or in very marshy ground adjacent to the lake and would have been susceptible to prolonged ice cover[35]. Another aquatic taxon, *Myriophyllum*, exhibited large peaks of abundance during MIS 5 at 111.7 and within MIS 7 at 201.5 ka with ~1700% and 700% the terrestrial pollen sum, respectively. Such abrupt oscillations of marginal aquatic taxa probably represented sudden changes in lake area, but they could equally represent a sudden expansion or contraction of the lake.

### Temperature, drought, and timing
The representation of upper montane forest species, which we take as being a proxy for warmth, was especially strong in MIS 15, 11 and 5e.

During these interglacials, *Podocarpus*, *Hedyosmum*, and *Weinmannia*, reached their peak abundances. Even during the less extreme events of MIS 7 and MIS 9, the proportion of upper montane species was similar to that of the Holocene (MIS 1). The only interglacial that was not marked by an increase in upper montane forest species record was MIS 13 (Fig. 3). That MIS 13 appears to have been unusually cool at this latitude is consistent with expectations based loess deposition and Antarctic temperatures[36].

During the most extreme warming observed in the Junín record, large changes in aquatic pollen and spores indicated rapid oscillations in lake level between high and low stands (Fig. 2). The rapid transition from a record rich in forest elements to one dominated by grasslands at c. 123 ka suggested a transition to drier conditions. The three long paleoecological records available from the Andes, those of Lakes Titicaca, Peru/Bolivia, Fúquene, Colombia, and Junín, all showed

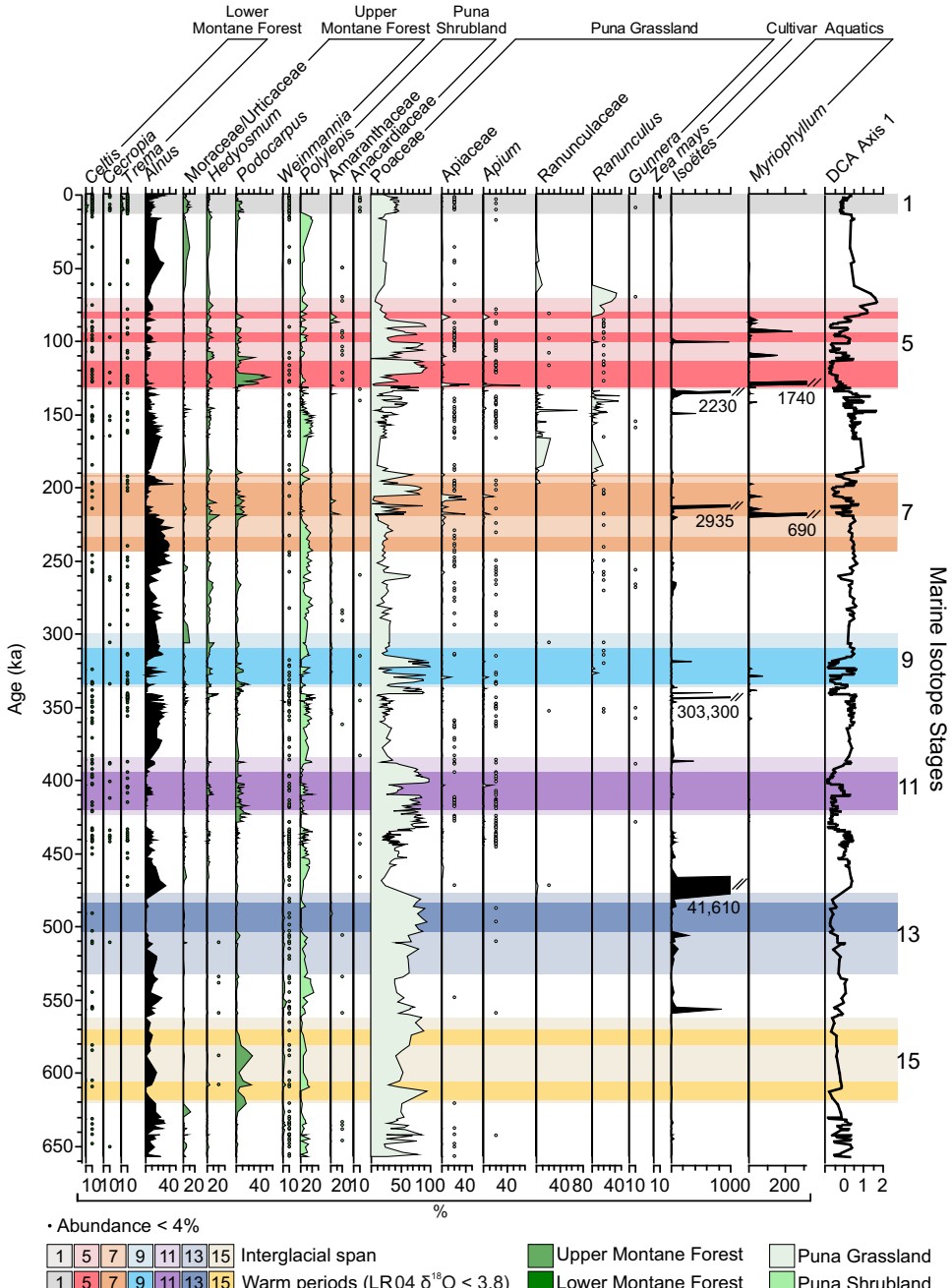

**Fig. 3 | Selected fossil pollen taxa from Lake Junín, Peru, showing vegetation changes across seven glacial-interglacial cycles.** Aquatic plants, Isöetes and *Alnus* are excluded from the pollen sum. Source data are provided as a Source Data file.

lowered lake levels during MIS5e (Fig. 4a–c). Of these records, Fúquene was the only setting that showed a complete change of biome (Fig. 4a–i) as it oscillated between interglacial forests and glacial grasslands[6]. Titicaca and the adjacent Altiplano showed the most pronounced changes in lake depth[37], and the greatest aridity as grasslands were replaced by a scrubby saltmarsh[7] (Fig. 4d–f). The ecological changes at Junín were less extreme than at either Fúquene or Titicaca as grassland elements persisted throughout, even though the warmest times supported forest species growing above their modern limit.

During the warm, wet phase of maximum forest expansion into the Junín Plateau, *Podocarpus* representation reached values of 59% of the pollen sum in MIS 5e (Fig. 3). Modern pollen studies revealed that *Podocarpus* is not strongly over-represented in pollen spectra (i.e. values > 1% are from nearby plants), and has a limited potential for

upslope dispersal[38,39]. *Podocarpus* may have colonized the warmest and most sheltered microhabitats close to the lake. Such locations may also have been microrefugia from fire[40]. Indeed, today it is uncommon to find *Podocarpus* above c. 3800 m elevation in Peru (Fig. 5), and the pollen abundances seen during past interglacials at Junín were exceptionally high.

To put this pollen abundance in perspective, of 93 modern pollen samples collected between 3000 m and 4550 m elevation above sea-level, including 11 from the Junín Plateau, none had >4% *Podocarpus* pollen, with a clear decline in representation above 3800 m[41] (Fig. 5). The same modern pollen dataset shows that *Hedyosmum and Weinmannia* are likely to be under-represented within their range and their pollen is not found above their actual range. In this data set, *Alnus* would appear to be similarly constrained, but the modern pollen were not collected at high enough elevations to see the over-representation

of *Alnus* in proglacial environments. It is important to note that the peaks of fossil abundance (Fig. 5) for *Podocarpus*, *Weinmannia* and *Hedyosmum* all occurred during interglacials, whereas the peak of *Alnus* abundance was during a glacial.

The early Holocene abundance of *Podocarpus* at Junín was about 15%, and if we take that as a baseline for a system with minimal human influence and near-modern climates, values > 15% in other interglacials could identify times when *Podocarpus* was closer to the lake or had larger populations near the lake (Fig. 3). *Podocarpus* appeared to arrive

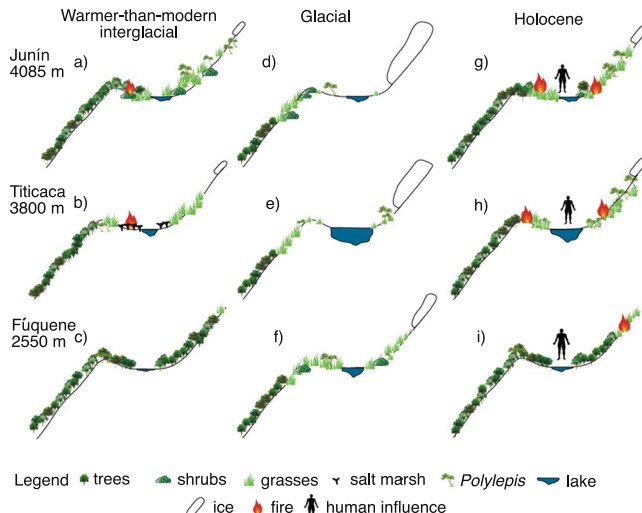

Junín
4085 m

Titicaca
3800 m

Fúqene
2550 m

Legend 🌲 trees   🌿 shrubs   🌾 grasses   ⌄ salt marsh   🌳 *Polylepis*   ◢ lake
       ⬭ ice   🔥 fire   🧍 human influence

**Fig. 4 | Schematic diagram illustrating vegetation and lake-level changes, and fire histories at the lakes Junín, Fúqene, and Titicaca.** Fúqene shows the most predictable biome response with glacial-interglacial transitions from Paramó to Andean forest. Junín shows some ecological variability but does not go through full biome transitions as seen at Fúqene and does not exhibit the ecological instability of Lake Titicaca. The arrival of people in the terminal Pleistocene influenced Holocene histories at all three sites. All icons courtesy of Nina Witteveen.

early in interglacials as the ice cover fell to local nadirs (Fig. 3). Moisture availability fluctuated between being low enough that peat accumulated (>90% organic matter) and high enough that a deeper lake formed in which carbonaceous muds were deposited[27]. As *Podocarpus* peaks occurred during both peat-rich and $CO_3$-rich layers precipitation variability was probably less important than warmth in determining the local population size of this tree (Fig. 6). On this basis all interglacials, with the exception of MIS 13, appear to have been as warm or warmer-than MIS 1.

The decline of *Podocarpus* in the mid-Holocene could have been caused by drier climates that were less favorable for *Podocarpus*, but there is no evidence for such drying in this record. Rather than invoking climate change, we consider it most likely that the increased use of the landscape by humans caused the apparent difference between modern (c. 1%) and early Holocene (15%) values (Fig. 6). Prior to human arrival, peaks of fire frequency needed high fuel availability, i.e. peaks of *Podocarpus*, and dry conditions indicated by high organic matter.

Detailed reconstructions of lake level reflect the relative strength of precipitation, stream inflows and outflows, evaporation, wind strength, and cloudiness[42]. Because Lake Junín is so shallow, millennial-scale oscillations between highstands and lowstands are apparent (Fig. 6)[29]. Overall, interglacials at Lake Junín showed increased moisture availability compared with full glacial conditions (Fig. 5), making this record more similar climatically to that of the High Plains of Bogotá, which had warm, wet, interglacials, rather than the warm, dry, ones of Lake Titicaca.

**Ecological trajectories and fire**

We analyzed the trajectories of all interglacials and compared them with that of MIS 1. The onset of each event was determined by a 5 x increase in pollen influx relative to the samples in the preceding glacial. These ecologically defined onsets were independent of the chronology, but strongly supported that age model. Where there were differences between the two estimates of interglacial initiation, the pollen data tended to lead the published age model by a few thousand years, i.e. within the uncertainty of the model (Table S3). We would note that

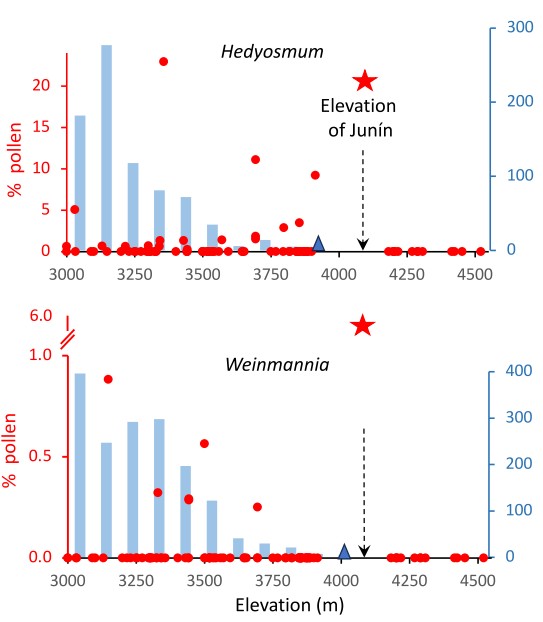

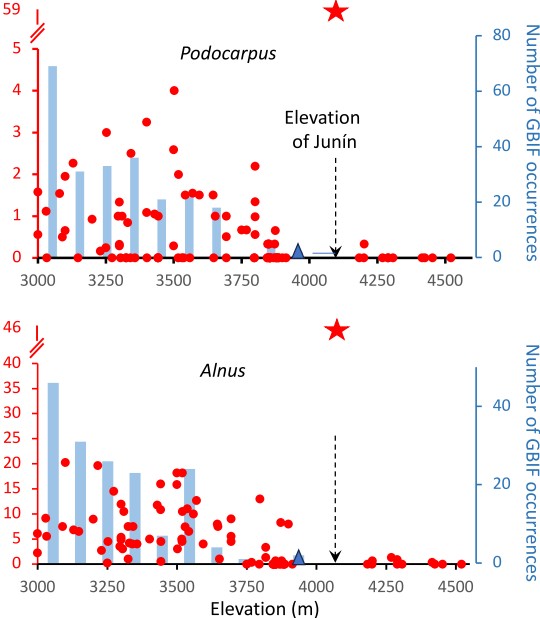

**Fig. 5 | Modern pollen abundance compared with documented plant occurrences across elevation for four Andean trees.** The occurrence of modern *Podocarpus, Hedyosmum, Weinmannia* and *Alnus* pollen in samples collected in Peru, Ecuador and Bolivia between 3000 and 4600 m elevation, source[41] relative to the number of occurrences per 100 m vertical increment. Source: Global Biodiversity Information Facility (GBIF.org)[79] accessed August 1st, 2023. Triangles mark the highest elevation record from GBIF. Stars indicate the highest pollen value from the JUN 15 core. Modern pollen data are based on *n* = 93 ecologically independent samples collected from Peru and Ecuador.

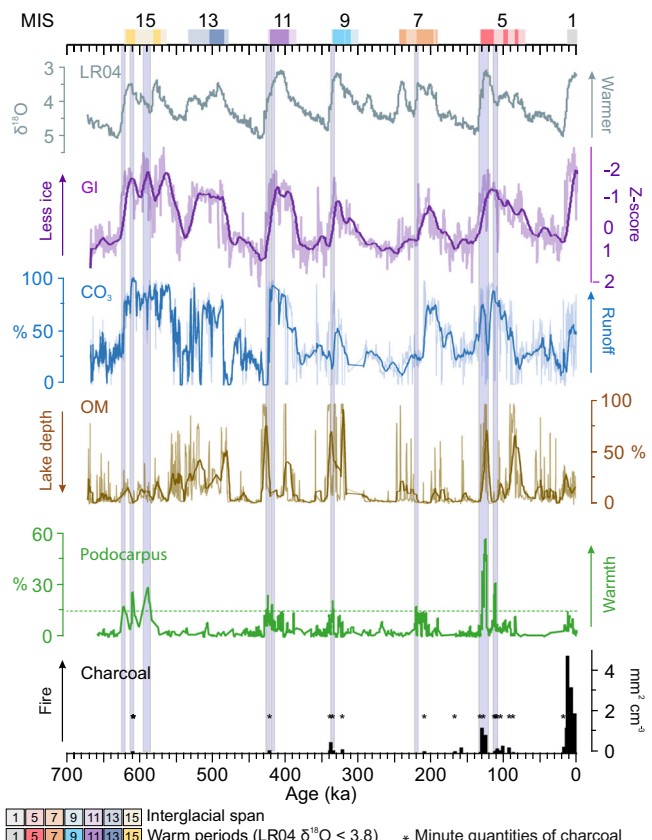

**Fig. 6 | Changes in paleoclimate proxies from Lake Junín, Peru, relative to changes in ocean temperature across seven glacial cycles.** The relative timing of environmental variables, peaks of charcoal, *Podocarpus* pollen, relative to the stacked ocean isotope record[77]. Interglacial peaks are marked by colored blocks as in Figs. 1, 2. GI Glacial Index a proxy for regional ice cover[26], OM Organic matter, high OM is interpreted as peat deposition in a shallow lake, Low $CO_3$ is a proxy for low runoff. Green dotted line is the maximum value for *Podocarpus* in MIS 1. Mauve bars highlight periods when *Podocarpus* in prior interglacials exceeded that of MIS 1. Source data are provided as a Source Data file.

MIS 7 had the weakest glacial-interglacial response and is the least reliably defined trajectory.

Because DCA maintains ecological distances between samples[43], by setting the DCA Axis 1 scores to zero at the onset of the interglacial, the subsequent trajectory of sample scores relative to that baseline provides an index of ecological distance, in this case increasing landscape productivity and forest influence. Compared with the other interglacials, MIS 1 is seen to have a relatively narrow range of ecological variability.

One of the strongest trends in the entire data set is the rarity of charcoal prior to c. 11.8 ka compared with a major increase in frequency and amount of charcoal in Holocene-aged samples (Figs. 2, 6). Sedimentary charcoal is an excellent proxy for fire[44], but it was absent in almost all glacial and interglacial samples. Prior to the Holocene, only the strongest interglacials, MIS 9 and 5e, revealed fire events in more than one sample (Table S1). In MIS 5e, the very warm and dry conditions induced burning and it is likely that fire played a part in limiting the spread of some woody species into the Junín Plateau (Fig. 4a). The higher peaks of pre-human charcoal in MIS 5e than in any other interglacial probably reflect maxima in three factors: temperatures, drought deficits, and fuel loads. Most Andean forest species are considered to be fire sensitive as they do not exhibit traits such as thick bark or resprouting capability[21,45]. *Podocarpus* and *Polylepis*, two of the most important arboreal elements in this record, are both fire-sensitive species[46], but can exist in fire-prone landscapes in fire-free microrefugia[40].

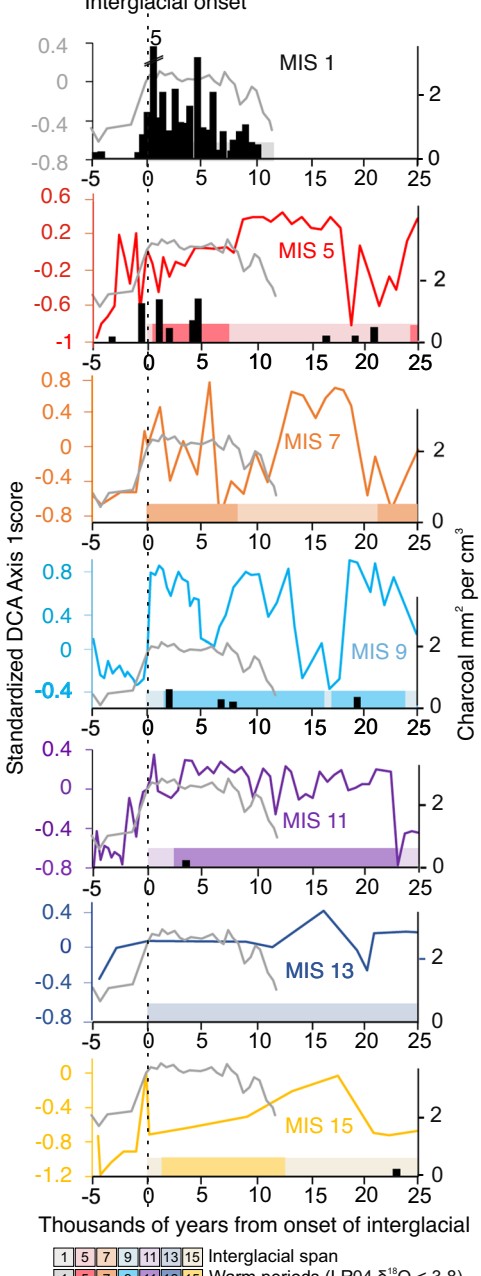

**Fig. 7 | The trajectory of vegetation changes at Lake Junín, Peru.** Charcoal and Detrended Correspondence Analysis (DCA) Axis 1 scores are plotted. DCA scores are calculated on the entire dataset and standardized by setting to zero at the start of each interglacial. Higher DCA values represent pollen assemblage changes toward landscapes with more arboreal elements and greater productivity. Plots run from 5000 years before the onset of the interglacial through the first 25,000 years of the interglacial. The Marine Isotope Stage (MIS) 1 curve is shown for comparison with each interglacial. Northern Hemispheric (primarily) warm periods, i.e. the LR04 benthic stack data, which is a proxy for deep sea temperature[77], are plotted for comparison. Source data are provided as a Source Data file.

As fires were almost exclusively limited to interglacials when productivity was relatively high, the times of greatest drying during glacials did not produce fire. Fuel appears to have been a more important constraint on fire activity at this elevation than drought. Despite the warmth early in MIS 5e, the high fuel availability, and periodic drought, the amount of fire, both in terms of frequency of detection (Table S1) and the amount of charcoal observed, was far less

than in MIS 1 (Fig. 7). MIS 1 stands out as having an almost ubiquitous charcoal presence in sediment samples. Regular fire activity is documented in a core recovered from the lake margin (core JU-96: Fig. 1) at c. 18 ka with a marked increase in charcoal amount at c. 13 ka, followed by an extirpation of megafaunal herbivores at c. 12.8 ka[14]. The data from core JUN 15, are similar with the first regular fire occurrence as early as 17.3 ka, and the increase in charcoal at c. 12.1 ka. The increase in fire activity observed in JUN 15 falls squarely in the emerging data from the tropical Andes of Late Pleistocene evidence for humans setting fires and eliminating megaherbivores c. 14–12 ka[11,12,14].

The loss of forest representation, which marks the end of interglacials, such as MIS 9 and 11, appears to be mimicked in the data for MIS 1, even though MIS 1 is not drawing to a close (Fig. 7). About 7000 years into MIS 1 (c. 4 ka) the downturn in DCA Axis 1 begins and the system appears to return to the openness of the late glacial state. Clearly, this is not a temperature driven response as the Andes are still 7–9 °C warmer than glacial temperatures[33]. Similarly, there is no evidence of a dramatic trend toward drier conditions over the last 4000 years, probably the opposite in fact[37]. Thus, we interpret this change in regional openness to reflect an acceleration of human clearance of forest and woodlands through burning and grazing[47], culminating in maize being grown near the lake c. 1.5–1 ka.

Such a trajectory fits well with available archaeological data[48,49]. Although people had started to initiate change in Andean landscapes as early as 14–12 ka and undoubtedly altered ecological trajectories[12], it was in the late Holocene that they produced manufactured landscapes through a combination of terracing, fire, camelid grazing, and crop cultivation[50,51].

## Tree line a product of manufactured landscapes?

The upper limit of continuous forest cover or tree line is often a fire-maintained rather than a physiologically driven boundary[21]. Prior to the transition towards a more open state in MIS 1, the pollen assemblage trajectories of MIS 1 and MIS 11 were very similar (Fig. 7); more similar than any of the other interglacials, as would be expected from the orbital parameters[10]. The vegetation of MIS 11 was always dominated by grasslands, but woodlands rich in *Polylepis* at the initiation and termination of the interglacial were replaced by ones rich in *Podocarpus, Hedyosmum*, and *Weinmannia* (Fig. 3). It was the loss of equivalent high-elevation woodlands in MIS 1 that gave the pollen signature its more open characteristic. While the trajectory of forest cover on the Junín Plateau was similar through the first 7000 years of the interglacials MIS 1 and 11, the next 4350 years were markedly different. The representation of woody taxa in MIS 11 pollen spectra continued to increase, and even though grasslands dominated, pockets of Andean forest became established. Presently, such pockets of Andean forest can be seen up to about 3800 m in Peru[45], while the more solid tree line often forms between 3400 and 3600 m elevation[52]. Human-induced deforestation can alter pollen representation to make the systems appear 'colder' and 'drier' as forests were replaced by grasslands[53]. Between c. 4350 cal BP and modern, we interpret MIS1 to have departed from the natural variability shown in MIS 11.

We found that during warmer-than-modern interglacials, woody taxa grew over a greater altitudinal range than is common today, but with upper distributional limits that did not co-occur, this may bring into question our modern conception of tree line. Tree line is characterized by high densities of stems and a transition from arboreal dominance to dominance by herbs, often grasses[54]. What we term tree line, which is often expressed as a sharp divide between grassland and forest, is a shifted baseline in what we accept as being natural, sensu[23,55]. Created and maintained by fire and grazing for millennia, the upslope members of arboreal populations have died out, creating a truncated distribution and a sharp boundary with the grassland. The natural state or pre-human state, in which fire would be so rare that it

does not structure habitats, is virtually unknown to us[16]. Our data suggest that the pre-human transition from forest to grassland across elevation was gradual with woody taxa extending far upslope with individual responses to environmental limits, not ending abruptly and uniformly on a line. Thus, while the Junín Plateau was dominated by grasslands throughout the last 670 ka, the lack of interglacial woody populations in the mid and late Holocene was a characteristic of a human-induced or manufactured landscape.

## Methods

### Site description

Lake Junín (11°01′S;76°07′W, 4085 m asl) lies within the Junín plateau in the central Peruvian Andes (Fig. 1). The lake is elongated north to south with an open-water surface area of c. 145 km². The basin is flat-bottomed and shallow with a maximum depth of 12 m[56]. Although moraine complexes flank the lake, the lake itself was not covered by glacial ice in the last 700,000 years (Rodbell et al., 2022). Much of the lake is bordered by dense sedge marshes comprised of *Scirpus totora*, *Scheonoplectus californicus*, and *Juncus* spp. with submerged macrophytes of *Chara*, *Myriophyllum*, and *Elodea* in the littoral zone[56–58]. Adjacent to the lake, the landscape is comprised of puna grassland, dominated by bunch (Ichu) grasses (e.g., *Festuca*, *Stipa*, and *Calamogrostis*) with some other common plants such as *Azorella*, *Plantago*, *Distichia*, and species of Asteraceae, Apiaceae, and Caryophyllaceae[59–61]. Shrubs (e.g., *Ephedra*, *Astragalus*, *Gynoxys*, and Ericaceae) and small trees (e.g., *Polylepis*) form isolated clumps far above the treeline. Downslope, at c. 3500–3700 m elevation, the puna transitions into upper Andean forests that are rich in *Weinmannia*, *Podocarpus*, *Hedyosmum*, *Myrsine*, *Alnus*, Asteraceae, Ericaceae, Rubiaceae, *Escallonia, Vallea*, and Lauraceae. Long-term management by grazing and fire has produced a manufactured landscape with unknown resemblance to its natural state[14].

Mean monthly temperature is 5–10 °C with a diurnal range that often exceeds 20 °C[62]. Seasonally wet, Lake Junín receives nearly 80% of the annual precipitation (-875 mm/yr) during the austral summer months (December through March)[63,64] and has a strong Atlantic influence[65]. Moisture is transported westward from the Atlantic Ocean and deposited in the high Andes through a combination of easterly trade winds and a succession of convective cells across Amazonia[66]. During the austral summer, tropical South Atlantic sea-surface temperatures increase and induce a southward migration of the Intertropical Convergence Zone (ITCZ). As the ITCZ migrates south the South American Low-Level Jet carries more Atlantic moisture into South America and stimulates development of the South American Summer Monsoon (SASM)[67].

### Field and laboratory methods

In 2015, an International Continental Scientific Drilling Program initiative raised long cores from Lake Junín. Prior publications have provided a detailed physical description of the composite sediment core Junín C15-1[68]. The chronology of this 88 m-long core was established using 80 [14]C ages from the upper 17 m, 12 U/Th-based age estimates from 53 U-Th dates, and 17 geomagnetic relative paleointensity tie point ages from the deeper sections (Fig. S2) (Hatfield et al. 2020b). The 110–670 ka portion of the age model was refined by Rodbell et al. (2022) by tuning physical property variations to the EPICA Dome C δD record using 10 tie points to improve alignment; almost all adjustments fell within the error envelope of the U-Th and RPI tuned age model.

Preparation of pollen samples ($n = 508$; volume = 0.5 cm³) followed standard protocols, including treatment with 10% HCl, 10% KOH, 10% $Na_4P_2O_7$, acetolysis, and sodium metatungstate flotation at a density between 2.0–2.1 g/mL[69]. Samples were spiked with a known quantity (-5000) of 15 μm polystyrene microspheres, to facilitate calculation of influx (grains per cm² per year).

Pollen samples were counted using a Zeiss Axioimager microscope at ×400 and ×630 magnifications. Pollen grains were identified using the Neotropical Pollen Database (Bush and Weng 2007), the Florida Institute of Technology pollen reference collection, and published pollen keys[33,70–72]. Pollen grains were recorded until either 300 terrestrial pollen grains were identified or 2000 microspheres were counted. For samples that contained >200 Poaceae grains, pollen identification continued until at least 100 non-Poaceae grains were identified or 1000 Poaceae grains were recorded. *Alnus* is a pioneer tree of disturbed forest edges well known to be massively over-represented in glacial forelands through being blown upslope and we exclude it from our Andean forest subtotals[33].

The fossil pollen data were analyzed using Detrended Correspondence Analysis (DCA) in the vegan[73] package for R[74]. To reduce noise in the analysis, only pollen types represented in 5 or more samples or occurring at least 2% of the pollen sum were included following[75]. Seventy-eight taxa met these criteria and were included in the analysis.

Additional sediment samples ($n = 705$; volume $= 0.5\,cm^3$) were placed in 3% $H_2O_2$ for 24 h to dissolve organic matter for charcoal analysis. Samples were then filtered through a 180 μm mesh. The resulting residue was placed in a petri dish of water and evaluated for charcoal using an Olympus stereoscope at ×20 and ×32 magnification. Surface area ($mm^2/cm^3$) of charcoal fragments were calculated using ImageJ software[76].

We take 11.8 ka to be the onset of the Holocene, but for all other interglacials we follow the temporal definitions of interglacial marine isotope stages (MIS) of[77]: MIS 1 11.8- 0 ka, MIS 5 130–115 ka, MIS 7 243–191 ka, MIS 9 337–300 ka, MIS 11 424–374 ka, MIST 13 533–478 ka, and MIS 15 621–563 ka. For the comparison of ecological trajectories during each interglacial, we define the onset of the interglacial functionally as the time that pollen influx (grains per $cm^2$ per yr) rose to five times the background amount of the preceding late glacial. The DCA Axis 1 score for the sample at the onset of the interglacial was set to zero and the trajectory of sample scores progressing into the glacial were plotted relative to that baseline.

Calculation of the glacial Index[26] is a *Z*-score of log-transformed sedimentary magnetic susceptibility and Ti/Ca data derived from X-ray fluorescence, at 250-year timesteps. The GI is the average *z*-score for both data sets. Organic matter and $CO_3$ were calculated based on coulometry of $CO_2$ released when samples were combusted at 1000 °C, for details see supporting on-line materials of[26].

### Reporting summary

Further information on research design is available in the Nature Portfolio Reporting Summary linked to this article.

## Data availability

The fossil pollen and charcoal data generated in this study have been deposited in the GITHUB database under accession code https://github.com/markbbush/Lake-Junin-data-files.git.

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

## Acknowledgements

We gratefully acknowledge the permission granted to us by the Peruvian government and especially officials of the Reserva Nacional de Junín for their assistance. We are indebted to the local communities around Lake Junín for allowing us to conduct this research. We thank the International Continental Scientific Drilling Program (ICDP) for their equipment and logistical support and DOSECC Exploration Services (USA) and Geotec (Perú) for drilling expertise. Blake Basler, Jennifer Berg, Shelby Conrad, Grace Harrington, Daniel Kenga, Madryn Larson, Genevieve Lucas, Jankiss Maher, James Ramos II, Emily Shrader, and Pilar Thomas, are thanked for their invaluable charcoal counting. This research was supported by grants from the ICDP (02-2012) and from the US National Science Foundation (EAR-1402054 to M.B., EAR-1402076 to D.T.R., EAR-1404113 to M.B.A., EAR-1404414 to D.M.)

## Author contributions

B.V., A.R.-D., M.A., D.R., conducted fieldwork, J.S. and M.K. conducted palynology, C.A. prepared figures, J.S. and M.B. wrote the first draft of the manuscript, C.C., A.W., R.G.H. and D.M. provided the chronology. M.B., J.S., B.V., A.R.-D., M.A., D.R., C.C., A.W., R.G.H. and D.M. contributed to writing the manuscript. D.R., M.A., M.B., and D.M. conceived the project.

## Competing interests

The authors declare no competing interests.
