## [Peer Review File · Nature Communications]

Reviewers' Comments:

Reviewer #1:

Remarks to the Author:

Nature Communications NCOMMS-23-26655-T peer-review

Schiferl, J, Kingston, M, Åkesson, CM, Valencia BG, Rozas-Davila A, McGee, D, Woods, A, Chen, CY, Hatfield, RG, Rodbell, DT, Abbott, MB, Bush, MB,, The uniqueness of the Holocene among interglacials: a neotropical perspective .

I am delighted to see a new most fascinating long continental pollen and charcoal record from the Peruvian Andes. Although providing unique windows into the Quaternary of terrestrial vegetation and climate change, such records are rare on a global scale (Hooghiemstra & Flantua 2022). I also see that making the pollen record available is lagging behind the records of the 'rapid' proxies, a characteristic we cannot prevent in absence of automated pollen analysis (Hooghiemstra 2023). My compliments to the members of this research project.

This paper is very welcome and of great relevance for a broad international audience. However, the paper has some flaws. The presentation of the setting, the data, and the conclusions needs an upgrade to make it understandable. Basic information of the altitudinal vegetation distribution (present-day interglacial, 'undisturbed' interglacial, and glacial setting) is missing. The pollen record itself is presented in 6 separate figures which makes a comparison between the records from different taxa, reflecting different ecological zones (elevational intervals), difficult. I certainly would like to see a 100%-wide column (main diagram) showing the changing proportions of puna grassland, puna shrubland, UMF, LMF (together making up the pollen sum), and aquatics in addition. The records showing the unidentified pollen taxa can be presented in a separate figure indeed. The present pollen record is too much compressed in depth (time) making a proper inspection of the pollen record impossible. Better to plot the complete pollen record (not cut into 6 pieces) with sufficient vertical space in order to see when exactly pollen taxa increase/decrease in the 6 glacial-interglacial cycles. This allows an exploration which taxa might be in competition within the vegetation zones (compositional change), and between the ecotones (e.g., reflecting altitudinal migration of the upper forest line, called 'tree line' in this paper). Taxa with low representation are shown in the 'fragile' pollen record with large black dots, therefore attracting more attention than deserved. Interesting is that several taxa make a start around 500 ka but this biostratigraphical zonation is not discussed. I would be interested if this jump in the composition of pollen spectra is real, or an effect of palynologist who analysed different intervals of the sediment 88-m core, or an effect of increasing familiarity with the regional pollen flora?

Specific comments:

32: add information about location and altitude.

33: explain 'empirically dated'

37: 'close to the lake?': I guess this record is able to provide a regional/Central Andes history of main vegetation dynamics and the inferred climate record.

29-42: The abstract is much ecology-focused while climate-related glacial-interglacial cycles, and basin related information is not mentioned.

52: while making reference to a site, always provide the altitude. Without that information in a mountain area, the reader is lost.

55: give altitude.

57-58: I think this is not a true contrasting history. While Bogotá is a 2550 m, halfway the glacial-interglacial amplitude of the upper forest line (UFL) (~2000 m min. to ~3500 m max.), Junin at 4085 m was always located above the UFL (during glacials and interglacials as well). Thus, forest-vs.páramo in Bogotá, and puna vs salt marsh in Junin is not a true comparison. By the way, in the Bogotá area also, intervals of salty marches occurred in dry rainshadow areas (e.g., Laguna la Herrera).

65-67: this claim is too wild. Indeed, in all vegetation zones human impact is significantly (Flantua et al. 2016) but 'unclear what the natural system would look like' is a bold exaggeration The older literature might be helpful for the authors; see for example Weberbauer 1911, Koepcke 1961, Young & Valencia 1992, Troll 1968, Cabrera 1968 in Troll ed. 1968.

72: the authors indeed clarified the term 'tree line' and use the term in harmony with the opinion of ecologists. However, in paleoecology it makes sense to avoid confusion. The upper boundary of

continuous Andean forest is the 'upper forest line' (UFL). Altitudinal positions of the UFL in the past can be reconstructed from the pollen record on the basis of AP% and taxonomic composition of the pollen spectra. The highest occurrence of individual trees /small patches of trees in the alpine grassy vegetation is the 'upper treeline'. The 'upper treeline' might be located up to 600 to 800 m above the UFL and, most important in paleoecology, the upper treeline cannot be reconstructed from a pollen record. In paleoecology it is helpful to make the difference.

77-80: here, some relevant references have been missed, and the claim in line 80 is not correct: see Moscol et al. 2009, 2010; Moscol & Cleef 2009a, 2009b; Jansen et al. 2013.

90-92: please provide a figure showing the altitudinal distribution of montane forest, puna, and glaciers during a full glacial, an optimum interglacial, and perhaps also the suggested distribution under current human impact.

94; can a landscape warm?

94: what is the meaning of 'productive'? Of course, montane forest was during interglacial conditions at closer distance to Lake Junin, but as far as I understand, Junin at 4085 m was never immersed in forest. See note 90-21. See Schmithüsen, who shows the UFL in Central Peru at ~3700 m.

95: I see no argument to claim 'productivity increased during interglacials'. We do not know anything about productivity of pollen. The source of poaceous pollen was at closer distance explaining higher influx values.

103-105: I see no convincing arguments for this claim, while it is not in harmony with the explanation in lines 105-106. During interglacial conditions montane forest was at closer distance to Lake Junin and therefore, montane forest is reflected by more pollen taxa in the interglacial pollen spectra. In the Bogotá record we showed that the taxonomic composition of montane forest during all interglacial periods of the last 1 million years was similar (Felde et al. 2016). I guess there is little reason to anticipate a different setting in Peru.

108: Has the literature about Peruvian uppermost montane forest composition been explored to see how exceptional this observation is? Possibly Malleux-Orjeda 1975, and ONERN 1976 might be helpful.

Figure 2: I am missing a 100%-wide column showing proportions of the categories graphed. Mind that cultivars and aquatics are not included in the pollen sum (the accolade below figure 2 suggests an erroneous composition or the pollen sum). Place an additional main diagram between DCE and MIS columns.

114-118: Which arboreal taxa are making up the ecotone forest in the Lake Junin area? What can be said about possible competition between these taxa?

119: give a reference for the 3800 m.

123: Fig S5 is insufficiently informative; it does not show the list of pollen taxa and the the %representation along the altitudinal gradient.

124-127: why is this discussion only focused on Podocarpus? I would like to see a similar discussion about montane forest in general. In fact, what I am missing in this paper is a reconstruction of the UFL position (and inferred temperature) over the full pollen record. Subsequently it can be specified for individual arboreal taxa, such as Podocarpus. So far, I have only seen the hand-waiving comparison between records shown in Fig. 1. Looking at 'Sky Islands a time travel through the Andes'

<https://www.youtube.com/watch?v=OWQOvLckL-g>
might give inspiration.

135-136: Podocarpus provides good timber and is among the first selected trees in deforestation (see Lake La Cocha in González-Carranza et al. 2012).

Figure 3: what means 0% and 100% lake depth? Or should the percentage bar be placed at the left?

Mind that the relationship 'deep water' during 'glacial conditions' was further explained in Van Boxel et al. 2016.

148: what means 'climatic pattern', can a more precise text be provided?

153: what are 'ecological trajectories'?

154: The term 'pattern' mostly refers to spatial data. In this paper 'pattern' is used for temporal data. I think it needs a clarification, or better, make a different word choice.

154-158: I am less surprised about the rarity of charcoal. The location at 4085 m locates Lake Junin in the dry puna during interglacials (~15% of the time reflected; Flantua et al. 2019), and below ice during most of the glacial intervals (perhaps over 50% of the time). We need a figure with crucial information for the central Andes as suggested in 90-92.

162-163: and what are the most arboreal taxa in the real vegetation?
165: under which conditions charcoal could be expected under glacial conditions? I am also confused here about 'productivity'. Could a clear sketch of the altitudinal vegetation distribution be provided followed by a discussion of the items as done here?
167-170: here, the natural setting (in MIS 5e) vs. a human occupation setting in MIS 1 is described. Is the evidence found not logical?
173-177: the JU C15-12 core evidence is interesting. Please could this relevant evidence be presented more transparently by making an additional figure?
178-180: I am surprised the authors find this evidence 'striking'. Dozens of pollen records show in the recent part of the Holocene signals of deforestation which equals to making the pollen record 'more glacial' [in a pollen record man behaves as an ice age].
181: what means 'system'?
181: 'Trend towards more open conditions': better to mention the changes that occurred in the ecosystems.
186: acceleration of human clearance: see the graphics in Mottl et al. 2022.
187: no evidence of potato cultivation? often coinciding with peaks of Rumex.
189 (195, 197, 203, 269, etc.): what is 'a trajectory of vegetation change'?
197: what is 'radically'? Please be careful in selling opinions (see also the papers by Moscol et al.). If so, it needs a radical explanation.
199: is 'thoroughly' hinting to species diversity, introduction of exotics, different altitudinal vegetation distributions, or, else?
201: mind the confusion in wording.
202: Unfortunately, there is little understanding what UFL migrations is driving physically. A mean annual temperature of c. 9.5°C in Colombia, and some degrees lower if the 'upper treeline' is in focus. It is an empirical relationship, hardly an understanding. Christian Körner (2012, 2021) is also discussing these uncertainties well in his books.
203: 'vegetation community' is a term used in descriptive phytosociology. Better to use here 'pollen assemblage'.
206-207: indeed, these are important arboreal taxa in the ecotone forest.
207-208: I would love to see the evidence in the pollen record. However, the records are presented highly compressed not allowing to explore the details discussed here.
214-215: see 178-180. Why also 'dryer'?
221: not documented in the pollen record?
221-222: I think the pollen record does not allow to make a conclusion about analogs.
223-224: the text 'this may bring into question our modern concept of tree line' is unclear and seems an unjustified statement. Please formulate carefully what idea is put forward here.
225: Here, I am not sure if I read the text well. To sharpen the discussion better to make difference between 'upper treeline' and 'upper forest line'.
227-229: The text 'Our data suggest that the transition from forest to grassland across elevation is far more gradual with woody taxa not ending abruptly' is puzzling. What is the message here? Studies of the vegetation structure across the UFL (e.g., Moscol & Cleef 2009a, 2009b) and dwarf trees entering the lower parts of the alpine zone and reaching even the higher elevations in the alpine zone (e.g., Polylepis) provides the UFL a rather abrupt, as well as a gradual aspect.
229-231: What is the clear message of this sentence? Both 'conclusions' are not surprising.
234-242: congratulations with this well developed age model.
258-259: for tropical forest this is quite a rigorous limitation in which much relevant information might be lost. I guess Birks & Gordon hinted more to pollen records from temperate areas.
Table S1 & Fig. S3: 'Dodonea' should read 'Dodonaea'.
Fig S3 (6 isolated parts): 'percentile' should read 'percentage'.
Please prepare a single full pollen diagram showing with a proper distance between the samples the taxa from puna grassland, puna shrub, UMF, LMF, and in addition the aquatics. Only in such way the diagram can be explored adequately and can support discussions and conclusions in the text.
Fig. S6: in which area were samples collected?

Supplementary Information

page 2, Figure S1: this map should move to the main text. It provides an excellent opportunity to add an elevational profile next to the map for 20 ka, 0 ka and, if rated of enough relevance, for the period just before human impact.

page 2: A mean annual temperature cannot be a range of 5°C.

In conclusion, the record presented is of amazing length and shows the last 7 glacial-interglacial cycles. The research team and authors are to be congratulated with the results.

The high elevation (4085 m) makes that Lake Junin was always immersed in tropical alpine puna vegetation. Montane forest was at close distance during interglacials (at c. 200-300? m vertical distance) and at c. 1500? vertical meters vertical distance during glacial maxima. But most of the time the UFL migrated somewhere in between these elevations. As a consequence, the UFL migrations have to be reconstructed 'from a few hundreds of meters distance' which might have complications (see Jansen et al. 2013). The resolution of the record is fair (670,000 yr / 508 samples makes 1318 yr mean resolution), possibly chosen for practical reasons.

In this paper I am missing a clear sketch of the altitudinal vegetation distribution at the various time intervals discussed. Remarkably enough, vegetation dynamics, and inferred climate change, during all glacial-interglacial cycles is hardly discussed. However, such design gives the paper a curious structure as we do not need a 670 ka record to make conclusions about the ecology of *Podocarpus* and to demonstrate the uniqueness of the Holocene in the series of Pleistocene interglacials only. Pollen records are interpreted in isolation, without a connection to the ever shifting altitudinal vegetation distribution in the Peruvian Andes; perhaps this approach was chosen in absence of sufficient understanding of the vegetation within the team of authors? (for several authors contributions were not specified).

I hope these comments are helpful for the authors to develop an improved draft of this paper about a most fascinating long continental pollen and charcoal record.

References cited:

- Flantua, S.G.A., Hooghiemstra, H., Vuille, M., Behling, H., Carson, J.F., Gosling, W.D., Hoyos, I., Ledru, M.P., Montoya, E., Mayle, F., Maldonado, A., Rull, V., Tonello, M.S., Whitney, B.S., Gonzalez-Arango, C., 2016, Climate variability and human impact in South America during the last 2000 years: synthesis and perspectives from pollen records, *Climate of the Past*, 12, 483-523, <https://doi.org/10.5194/cp-12-483-2016>.
- Flantua, S.G.A., Hooghiemstra, H., Vuille, M., Behling, H., Carson, J.F., Gosling, W.D., Hoyos, I., Ledru, M.P., Montoya, E., Mayle, F., Maldonado, A., Rull, V., Tonello, M.S., Whitney, B.S., Gonzalez-Arango, C., 2016, Climate variability and human impact in South America during the last 2000 years: synthesis and perspectives from pollen records.
- Flantua, S.G.A., O'Dea, A., Onstein, R., Giraldo, C., Hooghiemstra, H. (2019), The flickering connectivity system of the north Andean páramos *Journal of Biogeography* 46(8), 1808-1825, doi: 10.1111/jbi.13607.
- Felde, V.A., Hooghiemstra, H., Torres, V., Birks, H.J.B., 2016, Detecting patterns of change in a long pollen-stratigraphical sequence from Funza, Colombia; a comparison of new and traditional numerical approaches, *Review of Palaeobotany and Palynology*, 234, 94-109, doi:10.1016/j.revpalbo.2016.08.003.
- González-Carranza, Z., Hooghiemstra, H., Vélez, M.I., 2012, Major altitudinal shifts in Andean vegetation on the Amazonian flank show temporary loss of biota in the Holocene, *The Holocene*, 22, 1227-1241, <https://doi.org/10.1177/0959683612451183>.
- Hooghiemstra, H., Sarmiento Pérez, G., Torres Torres, V., Berrio, J-C., Lourens, L., Flantua, S.G.A., 2022. 60 years of scientific deep drilling in Colombia; the north Andean guide to the Quaternary. *Scientific Drilling*, 30, 1-15, <https://doi.org/10.5194/sd-3-1-2022>.
- Hooghiemstra, H., 2023, Making a long continental pollen record, a fabulous and bizarre enterprise: a 50-years retrospective. *Palynology* 2023: 1-5 (Letter to the Editor), <https://doi.org/10.1080/01916122.2023.2191257>.
- Jansen, B., De Boer, E.J., Cleef, A.M., Hooghiemstra, H., Moscol-Olivera, M., Tonneijck, F.H., Verstraten, J.M., 2013, Comparison of biomarker-based and pollen-based reconstructions of late Holocene forest dynamics in northern Ecuador, *Palaeogeography Palaeoclimatology Palaeoecology*, 386, 607-619, <https://doi.org/10.1016/j.palaeo.2013.06.027>.
- Koepcke, HW 1962. Synökologische Studien an der Westseite der peruanisches Anden. *Bonner Geographische Abhandlungen* 29, 320 pp.
- Körner C., *Alpine plant life; functional plant ecology of high mountain ecosystemns*. Springer, 500 pp.
- Körner, C., 2012. *Alpine treelines; functional ecology of the global high elevation tree limits*. Springer, 220 pp.

Moscol Olivera, M.C., Cleef, A.M., 2009a. Vegetation composition and altitudinal distribution of montane rain forests in northern Ecuador. *Phytocoenologia* 39, 175–204.

Moscol Olivera, M.C., Cleef, A.M., 2009b. A phytocoenological study of the páramo along two altitudinal transects in El Carchi province, northern Ecuador. *Phytocoenologia* 39, 79–107.

Moscol-Olivera, M., Duivenvoorden, J.F., Hooghiemstra, H., 2009. Pollen rain and pollen representation across a forest-páramo ecotone in northern Ecuador, *Review of Palaeobotany and Palynology*, 157, 285-300, <https://doi.org/10.1016/j.revpalbo,2009.05.008>.

Moscol-Olivera, M.C., Hooghiemstra, H., 2010. Three millennia upper forest line changes in northern Ecuador: pollen records and altitudinal vegetation distributions, *Review of Palaeobotany and Palynology*, 163(1), 113-126, <https://doi.org/10.1016/j.revpalbo.2010.10.003>.

Mottl, O., Flantua S.G.A., Bhatta, K.P., Felde, V.A., Giesecke, T., Goring, S., Grimm, E.C., Haberle, S., Hooghiemstra, H., Ivory, S., Kunes, P., Wolters, S., Seddon, A.W.R., Williams, J.W., 2021, Global acceleration in rates of vegetation change over the past 18,000 years, *Science*, 372, 860-864, doi:10.1126/science.abg1685.

Malleux-Orjeda, J., 1975. Mapa forestal del Peru (Memoria explicativa), Universidad Nacional Agraria La Molina, 161 pp.

ONERN 1976, Mapa ecologico del Peru, Guia Explicativa, Lima, 146 pp.

Schmithüsen, J., 1976. Atlas zur Biogeographie. Bibliographisches Institut Mannheim/Wien/Zürich, Band 3 Grosser Physischer Weltatlas.

Troll, C., 1968. Geo-ecology of the mountainous region of the tropical Americas. *Colloquium Geographicum*, 9, 223 pp. Ferd. Dummler Verlag, Bonn.

Van Boxel, J.H., González-Carranza Z., Hooghiemstra, H., Bierkens, M., Vélez, M.I., 2014, Reconstructing past precipitation from lake levels and inverse modelling for Andean lake La Cocha, *Journal of Palaeolimnology*, 51, 63-77, <https://doi.org/10.1007/s10933-013-9755-1>.

Young, KR, Valencia, N., 1992. Biogeografía, ecología y conservación de bosque montaño en el Peru. *Memorias del Museo de Historia Natural*, Lima, 221 pp.

Weberbauer, A., 1911. Die pflanzenwelt der peruanischen Anden, Verlag Wilhelm Engelmann, Leipzig / reprint Gantner Verlag, Vaduz355 pp.

Henry Hooghiemstra, Amsterdam, 5 July 2023.

Reviewer #2:

Remarks to the Author:

Reviewing this submission was a thoroughly enjoyable exercise, largely because the paper is well written and well structured and reports exciting new data.

The paper builds on the 2022 publication (Rodbell, D. T. et al. 700,000 years of tropical Andean glaciation. *Nature* 607, 301-306) that reported the chronology of this high altitude Andean site as well as a novel index for glaciation. Here, the palaeovegetation aspect of this project makes an important contribution to our understanding of ecological/landscape change under Late Pleistocene climate shifts in the tropics. The authors conclude that except for the most recent past, temperature has driven vegetation change on the Junin Plateau for the last 700,000 years. The current interglacial (the Holocene) covering approximately the last 12,000 years, stands out as different to all preceding interglacials, largely because of alterations to the landscape brought about by people and fire. The primary evidence for this resides in the charcoal record, as a proxy for fire.

The methods are well documented and the authors make the commitment to publish all data through NEOTOMA.

One quibble with the paper is that it opens with a summary (lines 49 -61) of how two other long records from the high Andes have contrasting interglacial histories. The High Plains of Bogota (9o N) and Lake Titicaca (18o S). The authors then go on to say that Lake Junín at 11o S and lying between the two will allow an investigation to evaluate if drought or temperature had the strongest effects on vegetation composition over the last 700,000 years. I realise it might be tricky to include some kind of comparison of these data with Lake Junin within the word and figure limits, but it might be more useful than Figure 2 (given the supplementary material). It would certainly

help support the statements within lines 148-151.

Other suggestions and corrections:

In the figure caption for Figure 1 "sediment organic content is listed" - see below. However, there is no sediment organic content plotted in Figure 1.

"Fig. 1: Summary pollen diagram from Lake Junín, Peru. Pollen are grouped according to modern habitat affinities relative to charcoal, sediment organic content, pollen influx and the LRO4 oceanic record (Lisiecki and Raymo 2005)."

There are a few things with Figure 3 I found a little confusing to read / interpret. For a start, CO3 is missing from the figure caption. There is also reference to 'Yellow bars' that aren't shown on the figure (see below). Would it be more sensible to have the GI index at the top of the figure, followed by CO3 plot and then the OM plot. Would some sort of smoothing put through the CO3 and OM plots (maybe in bold) and overlying the greyed out raw data make these two easier to read?

"Fig. 3: The relative timing of peaks of charcoal, Podocarpus pollen, and warm events. Interglacial peaks are marked by colored blocks as in Figs 1 and 2. GI = Glacial Index a proxy for regional ice cover 19, OM = Organic matter, high OM is interpreted as peat deposition in a shallow lake. Green dotted line is the maximum value for Podocarpus in MIS 1. Yellow bars highlight periods when Podocarpus in prior interglacials exceeded that of MIS 1."

Also, CO3 is noted as a proxy for runoff in Figure 3, and in the methods section we are referred to Rodbell, D. T. et al. (2022) for an explanation of this proxy as well as the methodology. However, they do not report CO3.

I would also suggest rewriting this sentence (lines 167-170) to make it easier to read and the meaning clearer.

"Despite the warmth early in MIS 5, the high fuel availability, and periodic drought, the amount of fire, both in terms of frequency of detection (Table S1) and the amount of charcoal observed, was far less than in MIS 1 (Fig. 3)."

REVIEWER COMMENTS

Reviewer #1 (Remarks to the Author):

Nature Communications NCOMMS-23-26655-T peer-review

Schiferl, J, Kingston, M, Åkesson, CM, Valencia BG, Rozas-Davila A, McGee, D, Woods, A, Chen, CY, Hatfield, RG, Rodbell, DT, Abbott, MB, Bush, MB,, The uniqueness of the Holocene among interglacials: a neotropical perspective .

I am delighted to see a new most fascinating long continental pollen and charcoal record from the Peruvian Andes. Although providing unique windows into the Quaternary of terrestrial vegetation and climate change, such records are rare on a global scale (Hooghiemstra & Flantua 2022). I also see that making the pollen record available is lagging behind the records of the 'rapid' proxies, a characteristic we cannot prevent in absence of automated pollen analysis (Hooghiemstra 2023). My compliments to the members of this research project.

This paper is very welcome and of great relevance for a broad international audience. However, the paper has some flaws. The presentation of the setting, the data, and the conclusions needs an upgrade to make it understandable. Basic information of the altitudinal vegetation distribution (present-day interglacial, 'undisturbed' interglacial, and glacial setting) is missing. The pollen record itself is presented in 6 separate figures which makes a comparison between the records from different taxa, reflecting different ecological zones (elevational intervals), difficult. I certainly would like to see a 100%-wide column (main diagram) showing the changing proportions of puna grassland, puna shrubland, UMF, LMF (together making up the pollen sum), and aquatics in addition.

We have modified Figure 1 to accommodate this request

The records showing the unidentified pollen taxa can be presented in a separate figure indeed. The present pollen record is too much compressed in depth (time) making a proper inspection of the pollen record impossible. Better to plot the complete pollen record (not cut into 6 pieces) with sufficient vertical space in order to see when exactly pollen taxa increase/decrease in the 6 glacial-interglacial cycles. This allows an exploration which taxa might be in competition within the vegetation zones (compositional change), and between the ecotones (e.g., reflecting altitudinal migration of the upper forest line, called 'tree line' in this paper). Taxa with low representation are shown in the 'fragile' pollen record with large black dots, therefore attracting more attention than deserved.

Interesting is that several taxa make a start around 500 ka but this biostratigraphical zonation is not discussed. I would be interested if this jump in the composition of pollen spectra is real, or an effect of palynologist who analysed different intervals of the sediment 88-m core, or an effect of increasing familiarity with the regional pollen flora?

Yes, there was a different analyst, Molly Kingston, for about half the counts in the MIS 13, 14, 15 and 16 sections. Jake Schifferl counted MIS 1-12 and half of MIS 13-16. We don't see a substantial difference between his counts and those of Molly Kingston. So we suspect that the apparent change is real in the sense it was not due to the counting, but it could also be a taphonomic change resulting from increasing lake size or the decreasing influence of the fringing marsh perhaps acting as a filter.

Specific comments:

32: add information about location and altitude.

Done

33: explain 'empirically dated'

Because of the 200-word limit on the abstract this is explained in methods.

37: 'close to the lake'? I guess this record is able to provide a regional/Central Andes history of main vegetation dynamics and the inferred climate record.

Edited to "grew within the Junín Plateau"

29-42: The abstract is much ecology-focused while climate-related glacial-interglacial cycles, and basin related information is not mentioned.

We have addressed this through a revision:

"Spanning seven glacial-interglacial oscillations, fossil pollen and charcoal recovered from the core provide evidence that the area around Lake Junín, Peru (4085 m elevation) was dominated by grasslands during both glacials and interglacials. Interglacials were characterized by higher productivity and an increase in woody taxa than glacials."

52: while making reference to a site, always provide the altitude. Without that information in a mountain area, the reader is lost.

Done

55: give altitude.

Done

57-58: I think this is not a true contrasting history. While Bogotá is a 2550 m, halfway the glacial-interglacial amplitude of the upper forest line (UFL) (~2000 m min. to ~3500 m max.), Junin at 4085 m was always located above the UFL (during glacials and interglacials as well). Thus, forest-vs.páramo in Bogotá, and puna vs salt marsh in Junin is not a true comparison. By the way, in the Bogotá area also, intervals of salty marches occurred in dry rainshadow areas (e.g., Laguna la Herrera).

Our point here is not that there are different vegetation types in the interglacial, but that the interglacials of Colombia were wet and the glacials drier, whereas the opposite was true of Titicaca.

We have attempted to clarify our point and relate it to the hypothesis of a hinge point raised by Bradbury (2001).

“In the forested setting of the high Plains around Bogota (5° N, 2540 m elevation), Colombia, cold, dry, grasslands replaced forests during glacial events, but each interglacial was marked by a temperature-driven upslope migration of forest species to produce assemblages like the forests of today. Conditions during interglacials appear to have been warm and wet. At Lake Titicaca (18°S, 3810 m elevation) in the Bolivian Altiplano, a temperature driven response of an upslope migration of forest was, during two major interglacials, MIS 5 and 9, interrupted by aridity. During both events, Andean forest migration stalled as the system transitioned to a saltmarsh, suggesting a drought-driven state¹. Consequently, these two sites show opposing patterns of when droughts peak, with the driest times in Colombia being during glacials versus interglacials at Titicaca. Bradbury ² suggested exactly this kind of climatic hinge point in the Andes, with locations north and south of c. 10 °S having opposing precipitation responses to glacial-interglacial cycles. At 11 °S, lying between Bolivia and Colombia, Lake Junín Peru, allows an investigation of the effects of interglacials of the last 700,000 years on vegetation composition to evaluate if drought or temperature had the strongest effects, and whether it conformed to the expectations of Bradbury ².”

65-67: this claim is too wild. Indeed, in all vegetation zones human impact is significantly (Flantua et al. 2016) but ‘unclear what the natural system would look like’ is a bold exaggeration The older literature might be helpful for the authors; see for example Weberbauer 1911, Koepcke 1961, Young & Valencia 1992, Troll 1968, Cabrera 1968 in Troll ed. 1968.

Our point here is that everything since about 12,000 BP is a shifted baseline, and we have little idea of what the high Andes would look like if there were still megafauna and a lack of fire and grazing. The older literature cited here is of limited use in looking at this as they were simply documenting a system that had already changed. The papers by Sylvester et al. have a very limited geographic scope but may document systems closer to what was natural, i.e. pre-human.

We have re-worded this:

“Subsequent camelid domestication, crop cultivation, and burning have transformed Andean landscapes to such a point that the Paramó grasslands became a manufactured landscape ³. Pre-human Paramós may have been richer in woody taxa than those of today ^{4,5} and had a softer boundary with Andean tree lines (the upper boundary of continuous Andean forest cover) ⁶.”

72: the authors indeed clarified the term ‘tree line’ and use the term in harmony with the opinion of ecologists. However, in paleoecology it makes sense to avoid confusion. The upper boundary of continuous Andean forest is the ‘upper forest line’ (UFL). Altitudinal positions of the UFL in the past can be reconstructed from the pollen record on the basis of AP% and taxonomic composition of the pollen spectra. The highest occurrence of individual trees /small

patches of trees in the alpine grassy vegetation is the 'upper treeline'. The 'upper treeline' might be located up to 600 to 800 m above the UFL and, most important in paleoecology, the upper treeline cannot be reconstructed from a pollen record. In paleoecology it is helpful to make the difference.

Yes, we agree and have tried to be clear throughout about this distinction.

77-80: here, some relevant references have been missed, and the claim in line 80 is not correct: see Moscol et al. 2009, 2010; Moscol & Cleef 2009a, 2009b; Jansen et al. 2013.

We are limited to the number of citations that we can add and so we have replaced both previously cited papers with Bakker et al. 2008 and Jensen et al 2013 and rephrased the sentence to reflect these findings:

"Prior paleoecological studies have attempted to track tree lines using fossil pollen, but evidence for tree line migration during the late Holocene found either no change ⁷ or a c. 200 m downslope displacement ⁸"

90-92: please provide a figure showing the altitudinal distribution of montane forest, puna, and glaciers during a full glacial, an optimum interglacial, and perhaps also the suggested distribution under current human impact.

We have added a figure (merged with reviewer 2's request) that offers a comparison of Titicaca, Fuquene and Junin under three sets of conditions: Glacial, MIS 1, and MIS 5e.

94; can a landscape warm?

We replaced landscape with environment.

94: what is the meaning of 'productive'? Of course, montane forest was during interglacial conditions at closer distance to Lake Junin, but as far as I understand, Junin at 4085 m was never immersed in forest. See note 90-21. See Schmithüsen, who shows the UFL in Central Peru at ~3700 m.

We are using productivity here to refer to landscape productivity. This is not just about individuals being nearer to the lake as the Poaceae were always there. But our interpretation is that the system has gone from scattered individuals (low landscape productivity) to a dense sward leading to a much more productive landscape - producing orders of magnitude more pollen - than those of a glacial foreland.

We have clarified that sentence to read:

"Peaks of pollen influx occurred during interglacial events as the environment warmed, the density of plants went from scattered individuals to a dense sward, and the landscape became more productive. Poaceae pollen influx (grains per cm² per yr) rose by two orders of magnitude as landscape productivity increased during interglacials."

95: I see no argument to claim 'productivity increased during interglacials'. We do not know

anything about productivity of pollen. The source of poaceous pollen was at closer distance explaining higher influx values.

Please see above.

103-105: I see no convincing arguments for this claim, while it is not in harmony with the explanation in lines 105-106. During interglacial conditions montane forest was at closer distance to Lake Junin and therefore, montane forest is reflected by more pollen taxa in the interglacial pollen spectra. In the Bogotá record we showed that the taxonomic composition of montane forest during all interglacial periods of the last 1 million years was similar (Felde et al. 2016). I guess there is little reason to anticipate a different setting in Peru.

We agree, but are unclear what is at issue here as this is our interpretation as well. We have modified the sentence to try to make this clearer.

“With the exception of MIS 13, interglacials were marked by an increase in upper montane forest species, e.g. *Podocarpus*, *Weinmannia*, and *Hedyosmum*, as they migrated upslope into the valley surrounding Lake Junín (Fig. 2).”

108: Has the literature about Peruvian uppermost montane forest composition been explored to see how exceptional this observation is? Possibly Malleux-Orjeda 1975, and ONERN 1976 might be helpful.

The suggested citations deal with biome boundaries and are not especially helpful here. We have extracted all 1168 records for *Podocarpus* for Peru and Ecuador from GBIF (i.e. approximately 10° N and S of the lake), and then took all those that occurred at >3000 m (237 individuals) and constructed a histogram of distributions at 100 m vertical intervals. *Podocarpus* are relatively abundant below 3400 m but only a few records get to 3800-3900 m elevation. The modern pollen representation tracks the shape of the curve of *Podocarpus* occurrence with no records > 2% in our database above 3400 m. Clearly, quite different from the ~60% *Podocarpus* pollen recorded at Junín in MIS 5e. We have added data for *Weinmannia*, *Alnus* and *Hedyosmum* for comparison.

Figure 2: I am missing a 100%-wide column showing proportions of the categories graphed. Mind that cultivars and aquatics are not included in the pollen sum (the accolade below figure 2 suggests an erroneous composition or the pollen sum). Place an additional main diagram between DCE and MIS columns.

This will now be shown in Fig. 1. Cultivars (1 grain) and aquatics were not included in any pollen sum.

114-118: Which arboreal taxa are making up the ecotone forest in the Lake Junin area? What can be said about possible competition between these taxa?

We can see *Podocarpus*, *Hedyosmum* and *Weinmannia* growing near the lake. I don't think we can say anything about competition.

119: give a reference for the 3800 m.

We have added the GBIF data to the figure that had been Fig.S6 and are bringing it into the main text as Fig. 4.

123: Fig S5 is insufficiently informative; it does not show the list of pollen taxa and the %representation along the altitudinal gradient.

Fig. S5 is the same DCA analysis, just highlighting the glacial as opposed to interglacial samples. We modified the caption to make this clearer.

124-127: why is this discussion only focused on Podocarpus? I would like to see a similar discussion about montane forest in general. In fact, what I am missing in this paper is a reconstruction of the UFL position (and inferred temperature) over the full pollen record. Subsequently it can be specified for individual arboreal taxa, such as Podocarpus. So far, I have only seen the hand-waiving comparison between records shown in Fig. 1. Looking at 'Sky Islands a time travel through the Andes'

<https://www.youtube.com/watch?v=OWQOvLckL-g>
might give inspiration.

I know the video well and we cannot make the inferences suggested based on this dataset. Unlike the Colombian records, the broader Andean forest never arrived at the lake and so we have no quantification of pollen abundances during forest presence.

We have added a section of text that describes the migration of taxa, the over-representation of *Alnus* and the peaks of aquatic taxa.

135-136: Podocarpus provides good timber and is among the first selected trees in deforestation (see Lake La Cocha in González-Carranza et al. 2012).

Yes, thank you for the citation, it has been incorporated.

Figure 3: what means 0% and 100% lake depth? Or should the percentage bar be placed at the left?

Placing the % bar to the left would clutter that margin. The arrow and lake depth are indicators of direction not of quantified measurement. We have reversed this subset of data to make it more intuitive with the rest of the figure.

Mind that the relationship 'deep water' during 'glacial conditions' was further explained in Van Boxel et al. 2016.

Thank you, we have incorporated this citation.

148: what means 'climatic pattern', can a more precise text be provided?

We have re-phrased this to be more explicit.

"Overall, interglacials at Lake Junín showed increased moisture availability compared with full glacial conditions (Fig. 3), making this record is more similar climatically to that of the High Plains of Bogota, which had warm, wet, interglacials, rather than the warm, dry, ones of Lake Titicaca."

153: what are 'ecological trajectories'?

We have defined it in a new topic sentence;

"Ecological trajectories are the responses to perturbations of a system. These trajectories can be viewed on many scales, including the replicability of millennial-scale responses to the onset of interglacials."

154: The term 'pattern' mostly refers to spatial data. In this paper 'pattern' is used for temporal data. I think it needs a clarification, or better, make a different word choice.

I have substituted 'trends' for 'pattern'.

154-158: I am less surprised about the rarity of charcoal. The location at 4085 m locates Lake Junin in the dry puna during interglacials (~15% of the time reflected; Flantua et al. 2019), and below ice during most of the glacial intervals (perhaps over 50% of the time).

We need a figure with crucial information for the central Andes as suggested in 90-92.

I agree that for whatever reason fire was absent during the glacials, and have added a note about this in text, but we note that much of the area around the lake was never ice-covered and have added an additional figure to the SOM to show this.

162-163: and what are the most arboreal taxa in the real vegetation?

As we state on line 171 "Most Andean forest species are considered to be fire sensitive".

165: under which conditions charcoal could be expected under glacial conditions? I am also confused here about 'productivity'. Could a clear sketch of the altitudinal vegetation distribution be provided followed by a discussion of the items as done here?

As stated above, I do not think we can 'reconstruct' beyond the environs of the Junin Plateau, but we have added a schematic figure. This plateau was not glaciated, but it is clear that glaciers surrounded the valley at some point, though we do not know the duration of ice cover.

167-170: here, the natural setting (in MIS 5e) vs. a human occupation setting in MIS 1 is described. Is the evidence found not logical?

Logical yes, but previously described? No, and given the local occurrence of woody species not really predictable at this elevation.

173-177: the JU C15-12 core evidence is interesting. Please could this relevant evidence be presented more transparently by making an additional figure?

These data have just been published in J. Biogeography⁹. Because one record is from the edge and the other from the middle of the lake they are only broadly similar records. Both records have about 50% grassland pollen, 5% lower montane forest pollen, and 30% upper montane forest, but individual peaks and troughs in the data don't align, and the edge core is far more detailed. The aquatic signature does not look similar. Getting into the taphonomic reasons why

these records should differ is a distraction for this paper and so I think it best not to go down that particular rabbit-hole.

178-180: I am surprised the authors find this evidence 'striking'. Dozens of pollen records show in the recent part of the Holocene signals of deforestation which equals to making the pollen record 'more glacial' [in a pollen record man behaves as an ice age].

I have deleted "striking" from that sentence.

181: what means 'system'?

Fixed

181: 'Trend towards more open conditions': better to mention the changes that occurred in the ecosystems.

Changed to "the downturn in DCA Axis 1".

186: acceleration of human clearance: see the graphics in Mottl et al. 2022.

I have added this citation

187: no evidence of potato cultivation? often coinciding with peaks of Rumex.

Not in this record.

189 (195, 197, 203, 269, etc.): what is 'a trajectory of vegetation change'?

See above

197: what is 'radically'? Please be careful in selling opinions (see also the papers by Moscol et al.). If so, it needs a radical explanation.

See below

199: is 'thoroughly' hinting to species diversity, introduction of exotics, different altitudinal vegetation distributions, or, else?

Reworded:

"...it was in the late Holocene that they produced manufactured landscapes through a combination of terracing, fire, camelid grazing, and crop cultivation^{10,11}."

201: mind the confusion in wording.

Text adjusted:

"The upper limit of continuous forest cover or tree line is often a fire..."

202: Unfortunately, there is little understanding what UFL migrations is driving physically. A mean annual temperature of c. 9.5°C in Colombia, and some degrees lower if the 'upper treeline' is in focus. It is an empirical relationship, hardly an understanding. Christian Körner (2012, 2021) is also discussing these uncertainties well in his books.

Yes, but we stand by the observation that this is a fire-maintained boundary in much of the Peruvian Andes.

203: 'vegetation community' is a term used in descriptive phytosociology. Better to use here 'pollen assemblage'.

Agreed

206-207: indeed, these are important arboreal taxa in the ecotone forest.

Agreed

207-208: I would love to see the evidence in the pollen record. However, the records are presented highly compressed not allowing to explore the details discussed here.

We are hoping that you like our less compressed records in text and the unified SOM Fig. S3 record can be expanded by the viewer.

214-215: see 178-180. Why also 'drier'?

Changes in microclimate caused by deforestation both increase temperature and decrease humidity. The consequence in pollen records is that we see a rise in Poaceae, which could be interpreted as either colder and/or drier as noted by Correa-Metrio et al.

221: not documented in the pollen record?

I have clarified this:

"However, as many other Andean taxa did not increase as markedly in abundance, i.e., *Alnus*, *Urticaceae*, *Ericaceae*, it seems these were habitats without modern analog sensu ¹²".

221-222: I think the pollen record does not allow to make a conclusion about analogs.

I disagree, even if only to note that our documentation of very high *Podocarpus* pollen in combination with a mixed grassland and montane forest assemblage is indicative of a no-analog flora.

223-224: the text 'this may bring into question our modern concept of tree line' is unclear and seems an unjustified statement. Please formulate carefully what idea is put forward here.

This has been reworded

225: Here, I am not sure if I read the text well. To sharpen the discussion better to make difference between 'upper treeline' and 'upper forest line'.

Please see below

227-229: The text 'Our data suggest that the transition from forest to grassland across elevation is far more gradual with woody taxa not ending abruptly' is puzzling. What is the message here? Studies of the vegetation structure across the UFL (e.g., Moscol & Cleef 2009a, 2009b) and dwarf trees entering the lower parts of the alpine zone and reaching even the higher elevations in the alpine zone (e.g., *Polylepis*) provides the UFL a rather abrupt, as well as a gradual aspect.

This is indeed complicated as it seems that in your definition of the upper forest line you would include *Polylepis*. *P. tarapacana* can grow up to the ice limit (5200 m elevation) and it and the other *Polylepis* with which it hybridizes are physiologically quite different from other trees. Hence, I am reluctant to use upper forest limit. I think the main point that we are trying to make is that the sharp boundary which almost always marks the edge of tree line (at least in Peru and Ecuador) is artificial.

We have reworded this paragraph to improve clarity:

“We found that during warmer-than-modern interglacials, woody taxa grew over a greater altitudinal range than is common today, but with upper distributional limits that did not co-occur, this may bring into question our modern conception of tree line. Tree line is characterized by high densities of stems and a transition from arboreal dominance to dominance by herbs, often grasses¹³. What we term tree line, which is often expressed as a sharp divide between grassland and forest, is a shifted baseline in what we accept as being natural sensu^{14,15}. Created and maintained by fire and grazing for millennia, the upslope members of arboreal populations have died out, creating a truncated distribution and a sharp boundary with the grassland. The natural state or pre-human state, in which fire would be so rare that it does not structure habitats, is virtually unknown to us⁴. Our data suggest that the pre-human transition from forest to grassland across elevation was gradual with woody taxa extending far upslope with individual responses to environmental limits, not ending abruptly and uniformly on a line. Thus, while the Junín Plateau was dominated by grasslands throughout the last 670 ka, the lack of interglacial woody populations in the mid and late Holocene was a characteristic of a human-induced or manufactured landscape.”

229-231: What is the clear message of this sentence? Both ‘conclusions’ are not surprising. We agree, but I think that is because, as paleoecologists, we are familiar with thinking about a pre-human state. It is clear to me that most neoecologists have not grasped this, and I think it is worthwhile to make these points.

234-242: congratulations with this well developed age model.
Thank you, it is undoubtedly a strength of this study.

258-259: for tropical forest this is quite a rigorous limitation in which much relevant information might be lost. I guess Birks & Gordon hinted more to pollen records from temperate areas.

DCA, is not particularly sensitive to the presence of rare taxa, but as with any multivariate analysis, if there is simply a lot of noise induced by including rare taxa it can degrade the analysis. Over the years, we have experimented with many permutations of the best abundances of taxa to include/exclude. To be honest, we find very little difference until you start excluding taxa at > 5% occurrence.

Table S1 & Fig. S3: ‘Dodonea’ should read ‘Dodonaea’.
Agreed

Fig S3 (6 isolated parts): ‘percentile’ should read ‘percentage’.

Agreed

Please prepare a single full pollen diagram showing with a proper distance between the samples the taxa from puna grassland, puna shrub, UMF, LMF, and in addition the aquatics. Only in such way the diagram can be explored adequately and can support discussions and conclusions in the text.

Please see new Fig. 1

Fig. S6: in which area were samples collected?

Peru, Ecuador and Bolivia, which has been added to the caption. Full details are in the cited source paper.

Supplementary Information

page 2, Figure S1: this map should move to the main text. It provides an excellent opportunity to add an elevational profile next to the map for 20 ka, 0 ka and, if rated of enough relevance, for the period just before human impact.

Done

page 2: A mean annual temperature cannot be a range of 5°C.

Yes, should have been 'monthly'

In conclusion, the record presented is of amazing length and shows the last 7 glacial-interglacial cycles. The research team and authors are to be congratulated with the results.

Thank you

The high elevation (4085 m) makes that Lake Junin was always immersed in tropical alpine puna vegetation. Montane forest was at close distance during interglacials (at c. 200-300? m vertical distance) and at c. 1500? vertical meters vertical distance during glacial maxima. But most of the time the UFL migrated somewhere in between these elevations. As a consequence, the UFL migrations have to be reconstructed 'from a few hundreds of meters distance' which might have complications (see Jansen et al. 2013). The resolution of the record is fair (670,000 yr / 508 samples makes 1318 yr mean resolution), possibly chosen for practical reasons.

The intent was always to sample interglacials much more extensively than glacials. The age model shifted many times over the years of counting, so we ended up with more glacial-aged samples than originally intended.

In this paper I am missing a clear sketch of the altitudinal vegetation distribution at the various time intervals discussed. Remarkably enough, vegetation dynamics, and inferred climate change, during all glacial-interglacial cycles is hardly discussed. However, such design gives the paper a curious structure as we do not need a 670 ka record to make conclusions about the ecology of Podocarpus and to demonstrate the uniqueness of the Holocene in the series of Pleistocene interglacials only.

Within constraints of location and journal space we have attempted to articulate what we believe to be the key points and have expanded our section on vegetation change and included a schematic diagram..

Pollen records are interpreted in isolation, without a connection to the ever shifting altitudinal vegetation distribution in the Peruvian Andes; perhaps this approach was chosen in absence of sufficient understanding of the vegetation within the team of authors? (for several authors contributions were not specified).

We have revisited author contributions

I hope these comments are helpful for the authors to develop an improved draft of this paper about a most fascinating long continental pollen and charcoal record.

References cited:

Flantua, S.G.A., Hooghiemstra, H., Vuille, M., Behling, H., Carson, J.F., Gosling, W.D., Hoyos, I., Ledru, M.P., Montoya, E., Mayle, F., Maldonado, A., Rull, V., Tonello, M.S., Whitney, B.S., Gonzalez-Arango, C., 2016, Climate variability and human impact in South America during the last 2000 years: synthesis and perspectives from pollen records, *Climate of the Past*, 12, 483-523, <https://doi.org/10.5194/cp-12-483-2016>.

Flantua, S.G.A., Hooghiemstra, H., Vuille, M., Behling, H., Carson, J.F., Gosling, W.D., Hoyos, I., Ledru, M.P., Montoya, E., Mayle, F., Maldonado, A., Rull, V., Tonello, M.S., Whitney, B.S., Gonzalez-Arango, C., 2016, Climate variability and human impact in South America during the last 2000 years: synthesis and perspectives from pollen records.

Flantua, S.G.A., O’Dea, A., Onstein, R., Giraldo, C., Hooghiemstra, H. (2019), The flickering connectivity system of the north Andean páramos *Journal of Biogeography* 46(8), 1808-1825, doi: 10.1111/jbi.13607.

Felde, V.A., Hooghiemstra, H., Torres, V., Birks, H.J.B., 2016, Detecting patterns of change in a long pollen-stratigraphical sequence from Funza, Colombia; a comparison of new and traditional numerical approaches, *Review of Palaeobotany and Palynology*, 234, 94-109, doi:10.1016/j.revpalbo.2016.08.003.

González-Carranza, Z., Hooghiemstra, H., Vélez, M.I., 2012, Major altitudinal shifts in Andean vegetation on the Amazonian flank show temporary loss of biota in the Holocene, *The Holocene*, 22, 1227-1241, <https://doi.org/10.1177/0959683612451183>.

Hooghiemstra, H., Sarmiento Pérez, G., Torres Torres, V., Berrio, J-C., Lourens, L., Flantua, S.G.A., 2022. 60 years of scientific deep drilling in Colombia; the north Andean guide to the Quaternary. *Scientific Drilling*, 30, 1-15, <https://doi.org/10.5194/sd-3-1-2022>.

Hooghiemstra, H., 2023, Making a long continental pollen record, a fabulous and bizarre enterprise: a 50-years retrospective. *Palynology* 2023: 1-5 (Letter to the Editor), <https://doi.org/10.1080/01916122.2023.2191257>.

Jansen, B., De Boer, E.J., Cleef, A.M., Hooghiemstra, H., Moscol-Olivera, M., Tonneijck, F.H., Verstraten, J.M., 2013, Comparison of biomarker-based and pollen-based reconstructions of late Holocene forest dynamics in northern Ecuador, *Palaeogeography Palaeoclimatology Palaeoecology*, 386, 607-619, <https://doi.org/10.1016/j.palaeo.2013.06.027>.

Koepcke, HW 1962. Synökologische Studien an der Westseite der peruanisches Anden. *Bonner*

Geographische Abhandlungen 29, 320 pp.

Körner C., Alpine plant life; functional plant ecology of high mountain ecosystems. Springer, 500 pp.

Körner, C., 2012. Alpine treelines; functional ecology of the global high elevation tree limits. Springer, 220 pp.

Moscol Olivera, M.C., Cleef, A.M., 2009a. Vegetation composition and altitudinal distribution of montane rain forests in northern Ecuador. *Phytocoenologia* 39, 175–204.

Moscol Olivera, M.C., Cleef, A.M., 2009b. A phytocoenological study of the páramo along two altitudinal transects in El Carchi province, northern Ecuador. *Phytocoenologia* 39, 79–107.

Moscol-Olivera, M., Duivenvoorden, J.F., Hooghiemstra, H., 2009, Pollen rain and pollen representation across a forest-páramo ecotone in northern Ecuador, *Review of Palaeobotany and Palynology*, 157, 285-300, <https://doi.org/10.1016/j.revpalbo.2009.05.008>.

Moscol-Olivera, M.C., Hooghiemstra, H., 2010, Three millennia upper forest line changes in northern Ecuador: pollen records and altitudinal vegetation distributions, *Review of Palaeobotany and Palynology*, 163(1), 113-126, <https://doi.org/10.1016/j.revpalbo.2010.10.003>.

Mottl, O., Flantua S.G.A., Bhatta, K.P., Felde, V.A., Giesecke, T., Goring, S., Grimm, E.C., Haberle, S., Hooghiemstra, H., Ivory, S., Kunes, P., Wolters, S., Seddon, A.W.R., Williams, J.W., 2021, Global acceleration in rates of vegetation change over the past 18,000 years, *Science*, 372, 860-864, doi:10.1126/science.abg1685.

Malleux-Orjeda, J., 1975. Mapa forestal del Peru (Memoria explicativa), Universidad Nacional Agraria La Molina, 161 pp.

ONERN 1976, Mapa ecologico del Peru, Guia Explicativa, Lima, 146 pp.

Schmithüsen, J., 1976. Atlas zur Biogeographie. Bibliographisches Institut Mannheim/Wien/Zürich, Band 3 Grosser Physischer Weltatlas.

Troll, C., 1968. Geo-ecology of the mountainous region of the tropical Americas. *Colloquium Geographicum*, 9, 223 pp. Ferd. Dummler Verlag, Bonn.

Van Boxel, J.H., González-Carranza Z., Hooghiemstra, H., Bierkens, M., Vélez, M.I., 2014, Reconstructing past precipitation from lake levels and inverse modelling for Andean lake La Cocha, *Journal of Palaeolimnology*, 51, 63-77, <https://doi.org/10.1007/s10933-013-9755-1>.

Young, KR, Valencia, N., 1992. Biogeografía, ecología y conservación de bosque montaña en el Peru. *Memorias del Museo de Historia Natural*, Lima, 221 pp.

Weberbauer, A., 1911. Die pflanzenwelt der peruanischen Anden, Verlag Wilhelm Engelmann, Leipzig / reprint Gantner Verlag, Vaduz 355 pp.

Henry Hooghiemstra, Amsterdam, 5 July 2023.

Reviewer #2 (Remarks to the Author):

Reviewing this submission was a thoroughly enjoyable exercise, largely because the paper is well written and well structured and reports exciting new data.

The paper builds on the 2022 publication (Rodbell, D. T. et al. 700,000 years of tropical Andean glaciation. Nature 607, 301-306) that reported the chronology of this high altitude Andean site as well as a novel index for glaciation. Here, the palaeovegetation aspect of this project makes an important contribution to our understanding of ecological/landscape change under Late Pleistocene climate shifts in the tropics. The authors conclude that except for the most recent past, temperature has driven vegetation change on the Junin Plateau for the last 700,000 years. The current interglacial (the Holocene) covering approximately the last 12,000 years, stands out as different to all preceding interglacials, largely because of alterations to the landscape brought about by people and fire. The primary evidence for this resides in the charcoal record, as a proxy for fire.

The methods are well documented and the authors make the commitment to publish all data through NEOTOMA.

One quibble with the paper is that it opens with a summary (lines 49 -61) of how two other long records from the high Andes have contrasting interglacial histories. The High Plains of Bogota (9o N) and Lake Titicaca (18o S). The authors then go on to say that Lake Junín at 11o S and lying between the two will allow an investigation to evaluate if drought or temperature had the strongest effects on vegetation composition over the last 700,000 years. I realise it might be tricky to include some kind of comparison of these data with Lake Junin within the word and figure limits, but it might be more useful than Figure 2 (given the supplementary material). It would certainly help support the statements within lines 148-151.

We have added a figure, trying to capture these ideas, though long time series are problematic so we have shown snapshots of Glacial, MIS 1 and MIS 5e vegetation.

Other suggestions and corrections:

In the figure caption for Figure 1 "sediment organic content is listed" - see below. However, there is no sediment organic content plotted in Figure 1.

"Fig. 1: Summary pollen diagram from Lake Junín, Peru. Pollen are grouped according to modern habitat affinities relative to charcoal, sediment organic content, pollen influx and the LRO4 oceanic record (Lisiecki and Raymo 2005)."

Agreed. Figure and caption have been revised.

There are a few things with Figure 3 I found a little confusing to read / interpret. For a start, CO3 is missing from the figure caption. There is also reference to 'Yellow bars' that aren't shown on the figure (see below). Would it be more sensible to have the GI index at the top of the figure, followed by CO3 plot and then the OM plot. Would some sort of smoothing put through the CO3 and OM plots (maybe in bold) and overlying the greyed out raw data make these two easier to read?

"Fig. 3: The relative timing of peaks of charcoal, Podocarpus pollen, and warm events. Interglacial peaks are marked by colored blocks as in Figs 1 and 2. GI = Glacial Index a proxy for

regional ice cover 19, OM = Organic matter, high OM is interpreted as peat deposition in a shallow lake. Green dotted line is the maximum value for Podocarpus in MIS 1. Yellow bars highlight periods when Podocarpus in prior interglacials exceeded that of MIS 1."

Fig 3 has been redrawn as per reviewer's suggestion and caption modified

Also, CO3 is noted as a proxy for runoff in Figure 3, and in the methods section we are referred to Rodbell, D. T. et al. (2022) for an explanation of this proxy as well as the methodology. However, they do not report CO3.

I would also suggest rewriting this sentence (lines 167-170) to make it easier to read and the meaning clearer.

"Despite the warmth early in MIS 5, the high fuel availability, and periodic drought, the amount of fire, both in terms of frequency of detection (Table S1) and the amount of charcoal observed, was far less than in MIS 1 (Fig. 3)."

- 1 Bush, M. B., Hanselman, J. A. & Gosling, W. D. Non-linear climate change and Andean feedbacks: An imminent turning point? *Global Change Biology* **16**, 3223-3232 (2010).
- 2 Bradbury, J. P., Grosjean, M., Stine, S. & Sylvestre, F. in *Interhemispheric Climate Linkages* (ed V. Markgraf) 265-291 (Academic Press, 2001).
- 3 Sarmiento, F. *Contesting Páramo: Critical Biogeography of the Northern Andean Highlands.*, (Kona Publishers, 2012).
- 4 Sylvester, S. P. *et al.* Relict high-Andean ecosystems challenge our concepts of naturalness and human impact. *Scientific Reports* **7** (2017).
- 5 Sylvester, S. P., Sylvester, M. D. & Kessler, M. Inaccessible ledges as refuges for the natural vegetation of the high Andes. *Journal of Vegetation Science* **25**, 1225-1234 (2014).
- 6 Sarmiento, F. O. & Frolich, L. M. Andean cloud forest tree lines: naturalness, agriculture and the human dimension. *Mountain Research and Development* **22**, 278-287 (2002).
- 7 Jansen, B. *et al.* Reconstruction of late Holocene forest dynamics in northern Ecuador from biomarkers and pollen in soil cores. *Palaeogeography, Palaeoclimatology, Palaeoecology* **386**, 607-619 (2013).
- 8 Bakker, J., Moscol Olivera, M. & Hooghiemstra, H. Holocene environmental change at the upper forest line in northern Ecuador. *The Holocene* **18**, 877-893 (2008).
- 9 Rozas-Davila, A., Rodbell, D. & Bush, M. B. Pleistocene megafaunal extinction in the grasslands of Junin-Peru. *Quaternary Research* **50**, 755-766 (2023).
- 10 Piperno, D. R. The origins of plant cultivation and domestication in the New World tropics: patterns, process, and new developments. *Current anthropology* **52**, S453-S470 (2011).

- 11 Mosblech, N. A. S., Chepstow-Lusty, A., Valencia, B. G. & Bush, M. B. Anthropogenic control of late-Holocene landscapes in the Cuzco region, Peru. *The Holocene* **22**, 1361-1372 (2012).
- 12 Overpeck, J. T., Webb, T. & Prentice, I. C. Quantitative interpretation of fossil pollen spectra: dissimilarity coefficients and the method of modern analogs. *Quaternary Research* **23**, 87-108 (1985).
- 13 Young, K. R. & León, B. Tree-line changes along the Andes: implications of spatial patterns and dynamics. *Philosophical Transactions of the Royal Society B*: **362**, 263-272 (2007).
- 14 Pauly, D. Anecdotes and the shifting baseline syndrome of fisheries. *Trends in Ecology and Evolution* **10**, 430 (1995).
- 15 Bush, M. B. *et al.* Fire and climate: contrasting pressures on tropical Andean timberline species. *Journal of Biogeography* **42**, 938-950 (2015).

Reviewers' Comments:

Reviewer #1:

Remarks to the Author:

Re-review Nature Communications, NCOMMS-23-26655-T: Schiferl, J, Kingston, M, Åkesson, CM, Valencia BG, Rozas-Davila A, McGee, D, Woods, A, Chen, CY, Hatfield, RG, Rodbell, DT, Abbott, MB, Bush, MB,, The uniqueness of the Holocene among interglacials: a neotropical perspective.

Dear editor,

Thank you for sending the improved draft of this exiting manuscript. It was enjoyable to read it again. I am fine to see how the authors have made changes and corrections on the basis of both review reports. The figures of the pollen records in particular improved substantially. Thanks to the authors. So, in my opinion this paper is almost ready for publication. I saw a few mistakes in the text and table.

54: in contrast to puna, páramo is a wet tropical alpine ecosystem. Delete 'dry'.

Figure 1: 'm.a.s.l.' should read 'm a.s.l.'

'Km' should read 'km' [following the metric system 'K' mean 'degrees Kelvin', while 'k' means 1000]

113: 'influxes' should read 'influx'.

140: 'pollen flora' should read 'pollen spectra'.

141: delete 'shrubby species'.

176: '7 and 9' should read 'MIS 7 and MIS 9'.

308: 'The MIS 11 pollen flora continued to incese' should read 'The MIS 11 diversity of the pollen spectra continued to increase'

Table S1: the following genera show an incorrect family name: Valeriana, Castilleja, Hydrocotyle, Myrsine, Rapanea, Sambucus, Celtis, and Trema.

Reference Cleef in the SI is incomplete. 'University of Amsterdam (1981)' should read 'Dissertationes Botanicae 61, 1-320 (1981)'.

With best regards,

Henry Hooghiemstra, Amsterdam, 2 October 2023.

Reviewer #2:

Remarks to the Author:

I have read through the revised manuscript and I am satisfied that the concerns and suggestions I raised in my initial review have been dealt with adequately.

Response to Reviewers

We have adopted all reviewer suggestions with the exception of changing family names in Table S2. The nomenclature that we adopt is that currently used by TROPICOS and includes recent reclassification of some genera into different families (leading to the discrepancy).